# Strain-Specific Therapeutic Potential of *Lactiplantibacillus plantarum*: A Systematic Scoping Review

**DOI:** 10.3390/nu17071165

**Published:** 2025-03-27

**Authors:** Oranut Chatsirisakul, Natasha Leenabanchong, Yada Siripaopradit, Chun-Wei Chang, Patsakorn Buhngamongkol, Krit Pongpirul

**Affiliations:** 1Faculty of Medicine, Chulalongkorn University, Rama IV Rd., Pathumwan, Bangkok 10330, Thailand; oranutchatsiri@gmail.com (O.C.); somyadasiripaopradit@gmail.com (Y.S.); putt12521@gmail.com (P.B.); 2Faculty of Medicine and Public Health, HRH Princess Srisavangavadhana College of Medicine, Chulabhorn Royal Academy, Lak Si, Bangkok 10210, Thailand; khuntasha@gmail.com; 3College of Medicine, National Taiwan University, Taipei 106319, Taiwan; b09401117@g.ntu.edu.tw; 4Center of Excellence in Preventive and Integrative Medicine, Faculty of Medicine, Chulalongkorn University, Rama IV Rd., Pathumwan, Bangkok 10330, Thailand; 5Department of Infection Biology & Microbiomes, Faculty of Health and Life Sciences, University of Liverpool, Liverpool L69 7ZX, UK; 6Bumrungrad International Hospital, Bangkok 10110, Thailand

**Keywords:** *Lactiplantibacillus plantarum*, probiotics, strain-specific effects, gut microbiome, immunomodulation, human clinical trials

## Abstract

**Objectives:** This systematically scoping review aims to evaluate the therapeutic potential and clinical benefits of specific *Lactiplantibacillus plantarum* (*L. plantarum*) strains in human health, identifying their strain-specific effects across various medical conditions. **Methods:** Following the PRISMA for Scoping Reviews (PRISMA-ScR) guidelines and employing the PICO framework, a comprehensive literature search was conducted in the PubMed and Embase databases to identify relevant studies published up to December 2023. Inclusion criteria were rigorously applied to ensure the selection of high-quality studies focusing on the clinical application of distinct *L. plantarum* stains. **Results:** This review analyzed several unique strains of *L. plantarum* across 69 studies, identifying several therapeutic benefits. *L. plantarum* 299v effectively improved gastrointestinal symptoms, enhanced oral health, and reduced systemic inflammation. *L. plantarum* IS-10506 exhibited notable immunomodulatory effects, especially in managing atopic dermatitis. *L. plantarum* LB931 showed promise in decreasing pathogenic colonization, supporting women’s vaginal health. Additionally, *L. plantarum* CCFM8724 demonstrated potential in reducing early childhood caries, highlighting its promise in pediatric oral care. **Conclusions:** The therapeutic potential of *L. plantarum* is extensive, with certain strains exhibiting promising clinical benefits for specific health concerns. The findings of this review advocate for the integration of *L. plantarum* strains into clinical practice, emphasizing the need for further research to elucidate their mechanisms of action, optimal dosages, and long-term safety profiles.

## 1. Introduction

Probiotics are live microorganisms that, when administered in adequate amounts, confer health benefits to the host [1]. They have been extensively studied due to their potential therapeutic roles in various clinical conditions, including immune disorders, cardiovascular diseases, hyperlipidemia, and hyperglycemia [2]. Among probiotic species, *Lactiplantibacillus plantarum* (*L. plantarum*), previously known as *Lactobacillus plantarum* [3], is recognized for its wide-ranging health benefits.

*Lactobacillus* species can be classified based on their fermentation capabilities into three functional groups: obligate homofermentative, obligate heterofermentative, and facultatively heterofermentative. Obligate homofermentative species, such as *L. salivarius* and *L. acidophilus*, produce primarily lactic acid. Obligate heterofermentative species, like *L. reuteri* and *L. fermentum*, generate lactic acid along with ethanol or acetic acid and carbon dioxide. Facultatively heterofermentative species, including *L. casei* and *L. plantarum*, produce lactic acid and carbon dioxide depending on environmental conditions [4,5,6,7].

*L. plantarum* is a Gram-positive lactic acid bacterium notable for its metabolic adaptability and ability to colonize diverse environments, such as fermented foods, meats, plants, and the gastrointestinal tract [8]. Its adaptability, environmental resilience, and extensive health-promoting effects distinguish it from other probiotics. Specific *L. plantarum* strains exhibit unique health-related functionalities. For instance, *L. plantarum* EGCG 13110402 demonstrates cholesterol-lowering effects [9], *L. plantarum* 299v improves vascular endothelial function [10], and *L. plantarum* HAC01 significantly reduces postprandial glucose and HbA1c levels, indicating potential benefits for individuals with prediabetes or type 2 diabetes [11].

The therapeutic potential of *L. plantarum* strains largely stems from their ability to modulate intestinal microbiota and immune responses, impacting overall health [12]. This strain-specific functionality is evident in clinical outcomes; *L. plantarum* 299v alleviates gastrointestinal inflammation by enhancing anti-inflammatory cytokines [13], *L. plantarum* PS128 supports neurological health and reduces oxidative stress in athletes [14,15,16], and *L. plantarum* Inducia positively influences metabolic and antioxidative responses related to cholesterol and BMI [17]. Thus, the precise selection of probiotic strains, similar to drug selection in pharmacology, is critical to effectively address specific clinical conditions, such as inflammatory bowel disease (IBD) [18] or neurological disorders [19].

In addition to therapeutic applications, *L. plantarum* is widely utilized in the food industry [3,20,21,22,23,24,25,26,27,28,29,30,31,32], particularly in fermented foods, due to its resilience to harsh conditions, low pH tolerance, and adherence to intestinal epithelial cells. These characteristics enhance its suitability for probiotic and therapeutic applications.

However, the clinical translation of *L. plantarum* benefits faces several limitations, including inconsistent study designs, small and heterogeneous samples, variable methodologies, and a lack of standardized outcome measures. Addressing these limitations through rigorous and targeted clinical investigations is essential for fully leveraging the therapeutic potential of *L. plantarum*. Therefore, bridging the gap between fundamental research and clinical application requires addressing these limitations through rigorous, targeted clinical investigations.

Therefore, this scoping review aims to address existing knowledge gaps by providing a comprehensive overview of current research on the therapeutic potential of individual *L. plantarum* strains across various health conditions. By summarizing strain-specific evidence, this review aims to inform future clinical guidelines, fostering the integration of *L. plantarum* into therapeutic or adjunct treatments and ultimately contributing to improved human health and well-being.

## 2. Materials and Methods

### 2.1. Registration of Protocols

This study was conducted in accordance with the Preferred Reporting Items for Systematic Reviews and Meta-Analyses extension for Scoping Review (PRISMA-ScR) guidelines. The protocol for this systematic review was retrospectively registered on INPLASY (INPLASY202520088) and is available on inplasy.com (accessed on 19 February 2025) (https://doi.org/10.37766/inplasy2025.2.0088).

### 2.2. Data Sources and Search Strategy

A systematic literature search was conducted using PubMed and Embase to identify relevant full-text articles published in English. The search strategy utilized the keyword “*Lactiplantibacillus plantarum”* or “*Lactobacillus plantarum*” or “*L. plantarum*” in “humans”. Additionally, the reference lists of included studies and relevant citations from other journals were screened via Google Scholar to identify additional studies.

### 2.3. Study Selection

A comprehensive literature search was conducted in two major biomedical databases, PubMed and Embase, to systematically identify peer-reviewed studies published up to December 2023. The search strategy involved targeted keyword combinations including “*Lactiplantibacillus plantarum*” or “*Lactobacillus plantarum*” or “*L. plantarum*” AND “humans”. Rigorous inclusion and exclusion criteria were applied during study selection, specifically selecting high-quality primary clinical studies involving distinct, clearly identifiable *L. plantarum* strains and reported therapeutic outcomes in human subjects. Screening was carried out in two stages (title/abstract screening and full-text screening) by at least two independent reviewers. Discrepancies were resolved through discussion and consensus or consultation with a third reviewer when necessary. Data extraction focused on strain-specific therapeutic outcomes, methodological quality, target population characteristics, study designs, and reported clinical endpoints. Data synthesis was qualitative, systematically summarizing findings to highlight strain-specific therapeutic benefits across identified health conditions.

Eligible studies included those that assessed *L. plantarum* as a monotherapy, without combining it with other probiotics species, in humans.

The study selection process involved four independent reviewers (OC, YS, CC, and PB), who screened articles for eligibility based on predefined inclusion and exclusion criteria. Any discrepancies between reviewers were resolved through group discussions in order to minimize the potential bias.

### 2.4. Data Extraction

Two independent reviewers (OC and NL) extracted data from the selected studies. The extracted information included the following: (1) study characteristics (*L. plantarum* strain, authors, year of publication, study type, source of publication, and country); (2) patient characteristics (sample size, type of disease, number of cases and controls); (3) outcomes (measurement methods for each disease and any additional relevant information).

All relevant text, tables, and figures in the included studies were reviewed for data extraction. The following types of studies were excluded: (1) non-original research (e.g., reviews, protocols, letters, comments, and guidelines); (2) studies that did not focus on *L. plantarum* monotherapy (i.e., mixing with other probiotic species); (3) non-human studies; (4) unpublished, gray literature or non-peer-reviewed studies; and (5) studies published in languages other than English in order to avoid any misunderstanding or language bias.

### 2.5. Data Synthesis and Analysis

The primary outcomes analyzed in this review included the following: the specific *L. plantarum* strains investigated and the types of diseases targeted by each strain. Since our study is a systematic scoping review, the checklist indicates that the risk of assessment bias is not liable to occur. The checklist can be found in the Appendix A.

### 2.6. Patient and Public Involvement

No patients or members of the public were involved in the design or conduct of this study. However, the findings of this systematic scoping review on the strain-specific therapeutic potential of *L. plantarum* could provide valuable insights for optimizing probiotic-based treatments for specific health conditions.

### 2.7. Methods, Research Design and PRISMA Flow Diagram

Our systematic scoping review was designed to comprehensively evaluate the therapeutic potential and clinical benefits of specific *L. plantarum* strains only in human health. The study followed the PRISMA for Scoping Reviews (PRISMA-ScR) guidelines, ensuring methodological rigor in the selection and synthesis of evidence. A structured approach was employed using the PICO framework to define the study parameters such as Population (study in humans only), Intervention (no species mixed of *L. plantarum* strains), Comparison (study compare between groups), and Outcomes (clinical benefits across various health conditions). A comprehensive literature search was conducted in PubMed and Embase to identify relevant studies published up to December 2023. In fact, it was our own limitation that meant that we did not review papers from all four databases in this study: Scopus, Embase, PubMed, and Cochrane library. In our study, the search strategy we used only obtained papers from the Embase and PubMed databases because they are very well-known and well-accepted databases in biomedical fields. Therefore, we decided to use only these two databases for our studies.

The inclusion criteria were established to select high-quality studies reporting primary data on human clinical outcomes related to specific *L. plantarum* strains. Following a systematic screening and selection process, data were extracted and qualitatively synthesized. The results were categorized by targeted health conditions, enabling a thorough assessment of therapeutic efficacy.

Strict inclusion and exclusion criteria ensured the selection of scientifically rigorous studies focused explicitly on distinct *L. plantarum* strains and their strain-specific clinical outcomes. This strategy minimized bias and enhanced the reliability of findings, supporting evidence-based clinical applications and future research.

Inter-rater reliability was assessed by having multiple reviewers independently screen studies based on predefined criteria. Discrepancies were resolved through consensus discussions or consultation with a third reviewer. This method ensured consistency, reduced subjective bias, and strengthened methodological integrity.

A total of 5481 studies were identified after a search across two databases. After excluding duplicates, 4142 studies’ titles and abstracts were screened. A total of 187 studies remained for full-text screening, and, finally, a total of 69 studies were included in our qualitative synthesis. The PRISMA 2020 flow diagram is shown in Figure 1.

## 3. Results

### 3.1. Data Characteristics Table: Please See Appendix A

This systematic scoping review covered a wide array of randomized, double-blind, placebo-controlled trials. The result of the study was comprehensively summarized in Appendix A. The study included sample sizes ranging from 20 to greater than 400. Healthy adults, children, older populations, and patients suffering from gastrointestinal, metabolic, anxious, or immune-related disorders were tested within a subject sample composed of a wide variety of ages, sexes, and geographies. Treatments ranged from about a week to a total of six months, with most intervention periods being within four to twelve weeks. The study dosages had high heterogeneity, with some specifying concentrations while others did not. The observed effects were markedly different by strain and clinical target. Some strains significantly lower cholesterol, relieve anxiety, improve bowel function, and modulate immunity, while others report very little or no significant clinical outcomes when compared with their placebo counterparts. Methodological, participant background, intervention period, and outcome variability among the studies further emphasize the need for some definite probiotic strain to allow the reliable conclusion and application of the results.

Table 1 includes an overall classification of the symptoms with reference numbers. The gastrointestinal system was classified into five major symptoms such as IBS symptoms, bowel functions, microbiome diversity, diarrhea, and constipation. For IBS symptoms, there are seven studies included in this study. There were classified into seven major symptoms for the immune system, three symptoms for the central nervous system, two symptoms for the integumentary system, and three symptoms for the miscellaneous group.

The graph of the *L. plantarum* strains classified by symptoms is also shown in Figure 2. The sunburst graph clearly presents the classification of *L. plantarum* strains by symptoms. Figure 2 shows that for infections under the subset of the immune system, there are several *L. plantarum* strains that help to alleviate infections, such as *L. plantarum* P17630 (two studies), *L. plantarum* DR7 (two studies), *L. plantarum* I1001, and *L. plantarum* PCS26. Therefore, this graph significantly helps readers to have an overall understanding of the studies at a glance.

The table of results, also shown in Table 2, are summarized from the Appendix A. From Table 2, *L. plantarum* CCFM1143 eases the symptoms of chronic diarrhea, while *L. plantarum* DR7 helps to reduce anxiety and improve cognition and URTI symptoms. However, *L. plantarum* 299v results in mixed results; some studies showed positive and some showed negative effects. There may be several related factors, but confounding factors may play a major role in this issue such as diet, age, genetic differences, and gut microbiome diversity, which may change the response to treatment; these have not been studied in this research.

### 3.2. Gastrointestinal System

#### 3.2.1. Irritable Bowel Syndrome (IBS)

Several *L. plantarum* strains have been investigated for their effects on IBS, yielding mixed outcomes. *L. plantarum* Lpla33 (DM34428) [37] and *L. plantarum* APsulloc 331261 [35] demonstrated significant efficacy in improving diarrhea-related symptoms in IBS patients, offering potential relief. For Lpla33, SCFAs, particularly butyrate, may help to promote intestinal barrier function, a feature of several *L. plantarum* strains. Bile acid (BA) metabolism may also be implicated, as *L*. *plantarum* Lpla33 possesses significant bile salt hydrolase activity. Diarrhea and visceral hypersensitivity have been associated with the decreased 7α-dehydroxylation of primary BAs to secondary BAs. Additionally, the ability of *L. plantarum* Lpla33 to modulate intestinal barrier function, inhibit key pathogens, and moderate inflammatory markers may have played a key role in these observed effects.

In contrast, *L. plantarum* 299v [33,38] showed no significant effect on abdominal pain, bloating, or rectal emptying when compared to a placebo. However, some studies noted modest improvements in abdominal pain and bloating [34]. The IBS symptoms were evaluated using a composite score [39]; *L. plantarum* 299v failed to show significant efficacy. The IBS symptom scores were virtually unchanged in each subject. Fecal weights did not change significantly, and total hydrogen production over 24 h was not reduced. This suggests that, in this case, *L. plantarum 299V* does reduce hydrogen production in the colon, but not sufficiently enough in this group of patients to be clinically effective.

*L. plantarum* MF1298 [59] was associated with unfavorable effects on IBS symptoms, indicating potential strain-specific variability in therapeutic outcomes.

#### 3.2.2. Bowel Function

The impact of *L. plantarum* strains on bowel function varies depending on the condition. In patients undergoing endoscopic sclerotherapy for internal hemorrhoids, *L. plantarum* MH-301 [49] demonstrated a positive effect on bowel movements. The oral *L. Plantarum* MH-301 has a beneficial effect on the improvement of both postoperative internal hemorrhoids and difficult defecation. It increased the relative abundance of Firmicutes at the phylum level. Firmicutes are mostly Gram-positive bacteria and are the dominant phylum in human intestinal microbiota. Firmicutes are health-promoting probiotics, including those such as *Lactobacillus* and *Ruminococcus*, which play a vital role in promoting health.

However, *L. plantarum* CJLP243 [48] did not improve bowel function in rectal cancer patients after ileostomy, suggesting that strain-specific benefits may not extend to all gastrointestinal conditions.

#### 3.2.3. Microbiome Diversity

Several *L. plantarum* strains have shown their ability to modulate gut microbiota composition. In children, *L. plantarum* 299v [58] significantly enhanced microbiome diversity. *L. plantarum* DR7 [61] maintained a balanced gut bacterial profile, supporting gut homeostasis with Bacteroidia and Bacteroidales, which were maintained upon the administration of DR7. It reduced plasma cortisol levels and exerted changes along the pathways of two neurotransmitters, namely, serotonin and dopamine. Although changes in gut microbiota upon the administration of DR7 was consistent along different taxonomic levels, the correlations of the changes in these microbial groups with changes in neurotransmitter gene expression also showed such consistency.

*L. plantarum* P-8 [60] increased the richness of beneficial bacteria and reduced the presence of opportunistic pathogens [63]. However, *L. plantarum* MF1298 [59] had no significant effects on gut microbiota composition or gut wall health. *L. plantarum* Q180 [62] was able to maintain a healthy intestinal environment in postprandial conditions, reinforcing its role in digestive health. The only LDL-cholesterol and ApoB-100 levels were decreased by *L. plantarum* Q180 supplementation for 12 weeks; the postprandial blood levels of lipid markers such as TG, ApoB-48, and ApoB-100 were also significantly decreased. It also helped to maintain healthy postprandial lipid metabolisms in subjects with a higher level of enteric bacteria such as *R. bromii*, *K. alysoides*, *B. intestinihominis*, and *F. plautii*.

#### 3.2.4. Diarrhea

The effects of *L. plantarum* strains on diarrhea varied across different populations and conditions. *L. plantarum* 299v [67] did not show beneficial effects in treating pediatric diarrhea, indicating limited efficacy for children. *L. plantarum* CCFM1143 [83] was effective in treating chronic diarrhea in adults, suggesting its potential for managing diarrhea in older populations.

*L. plantarum* LRCC5310 [82] significantly improved rotavirus-induced diarrhea in children, demonstrating its promise in treating infectious diarrhea in young patients. Rotavirus is one of the most common causes of Acute gastroenteritis (AGE) in children. Rotavirus consists of a segmented gene and can represent a variety of genes by a combination of G-P proteins. Generally, the genotypes of viruses that cause AGE in children are typically G1P, G2P, G3P, etc. Moreover, the intake of LRCC5310 was found to be effective in the suppression of viral symptoms as well as in prognosis and treatment via virus titer reduction.

#### 3.2.5. Constipation

Several *L. plantarum* strains have demonstrated efficacy in relieving constipation. *L. plantarum* Lp3a [89] effectively improved functional constipation, making it a potential therapeutic option. Functional constipation (FC) is a common gastrointestinal (GI) disorder characterized by symptoms of constipation without a clear physiologic or anatomic cause. Gut microbiome dysbiosis has been postulated to be a factor in the development of FC, and treatment with probiotic regimens, including strains of *Lactobacillus plantarum (L. plantarum)*, has demonstrated efficacy in managing symptoms. Lp3a relieves FC symptoms by enhancing intestinal motility and putatively modulating methane metabolism and bile acid synthesis. The bile acids serve as physiological laxatives; this suggests a role for the modulation of bile acid metabolism as an additional putative mechanism of Lp3a in FC.

*L. plantarum* IS10506 [87] specifically helped in treating functional constipation in women, suggesting gender-specific benefits. *L. plantarum* P9 [87,88] consistently relieved chronic constipation across studies, reinforcing its potential as a reliable treatment for this condition.

### 3.3. Immune System

#### 3.3.1. Immune Response

*L. plantarum* IS-10506 [40] has been linked to an increase in secretory immunoglobulin A (sIgA) in children, a key component of mucosal immunity. This enhancement may improve the body’s ability to respond to pathogens and allergens. *Lactobacillus plantarum* IS-10506 supplementation significantly increased fecal secretory immunoglobulin A (sIgA) levels in children under two years old. The probiotic stimulates transforming growth factor-β1 (TGF-β1), which plays a crucial role in regulating immune responses and promoting sIgA production. The study found a strong correlation between higher TGF-β1/TNF-α ratios and increased sIgA levels, suggesting that IS-10506 enhances mucosal immunity, helping to protect against infections.

#### 3.3.2. Infections

Probiotics have demonstrated potential in managing infections, including upper respiratory tract infections (URTI), urinary tract infections (UTI), and vulvovaginal candidiasis (VVC). *L. plantarum* DR7 [50,51] significantly reduced the severity and duration of URTI symptoms, indicating its potential as an immune-boosting probiotic.

*L. plantatum* PCS26 [87], *Lp*26, was found to alleviate UTI symptoms, suggesting it could serve as an alternative or adjunct to antimicrobial treatments. Urinary tract infections (UTI) are frequent bacterial infections in childhood. *L. plantarum* 26 was derived from local Slovenian cheese in the Pathogen Combat Project. Lp26 helps to relieve UTI symptoms primarily through its antimicrobial effect against E. coli, potentially reducing pathogenic bacteria in the gut and their migration to the urinary tract.

*L. plantarum* P17630 [52] was effective in reducing vaginal discomfort and preventing recurrent VVC infections [55]. *L. plantarum* I1001 [54] was identified as a compound that enhances the efficacy of treatment for recurrent VVC.

#### 3.3.3. Atopic Dermatitis

Probiotics have shown promise in alleviating symptoms of atopic dermatitis, a chronic inflammatory skin condition. *L. plantarum* IS-10506 [66] demonstrated significant reduction in both adults and children [67], likely due to its immunomodulatory effects.

*L. plantarum* CJLP133 [65] also showed beneficial effects on atopic dermatitis in children [59], making it a promising probiotic for managing skin inflammation by modulating immune responses. In children with moderate to severe AD, those who responded well to probiotic treatment had high total IgE levels, increased transforming growth factor-beta (TGF-β) expression, and a high proportion of CD4+CD25+Foxp3+ regulatory T (Treg) cells. The supplementation of *L. plantarum* CJLP133 showed significant reductions in AD severity and an increase in Treg cells, indicating improved immune regulation.

#### 3.3.4. Cancer

In cancer care, probiotics have been studied for their role in supporting gastrointestinal health and nutritional status. *L. plantarum* 299 [84] alleviated gastrointestinal symptoms in cancer patients, potentially improving their quality of life during chemotherapy or radiation treatments. *L. plantarum* 299v [85] demonstrated potential to improve the nutritional status of cancer patients, supporting overall health and recovery during cancer treatment. *L. plantarum* 299v (Lp299v) demonstrated potential to improve the nutritional status of cancer patients receiving home enteral nutrition (HEN) by preventing weight loss and enhancing nutrient absorption. It helps to maintain body weight and muscle mass, crucial for preventing cancer cachexia. *L. plantarum* 299v increased serum albumin, total protein, and transferrin levels, suggesting improved protein metabolism and reduced inflammation. It also modulated gut microbiota by increasing beneficial bacteria and enhancing mucin production (MUC2 and MUC3), which strengthen gut barrier function.

#### 3.3.5. Human Immunodeficiency Virus (HIV)

The immune-modulating properties of *L. plantarum* have been examined in HIV-infected individuals with mixed results. *L. plantarum* IS-10506 [12] did not show a significant effect on overall immune responses in HIV-infected children. *L. plantarum* IS-10506 is a probiotic strain isolated from Indonesian fermented buffalo milk (dadih) with demonstrated gut barrier-enhancing properties. It has been shown to mitigate intestinal epithelial damage by inhibiting nuclear factor kappa B (NF-κB) activation and downregulating tumor necrosis factor receptor-1 (TNFR1), which play key roles in inflammatory responses. It significantly reduced blood lipopolysaccharide (LPS) levels, indicating reduced microbial translocation and improved gut integrity in HIV-infected children undergoing antiretroviral therapy (ARV). However, its effect on systemic and humoral immune responses, including CD4+/CD8+ T cell ratios and fecal secretory IgA (sIgA) levels, was not significant. It increased the number of regulatory T cells [90] in children receiving first-line antiretroviral therapy (ART), suggesting a potential role in immune regulation for managing HIV-related immune dysregulation.

#### 3.3.6. Vaginal Health

*L. plantarum* LB931 [91] was observed to maintain a low vaginal pH, which helps to prevent harmful microbial overgrowth. This contributes to reducing the risk of bacterial vaginosis and yeast infections, promoting overall vaginal health. It helps to maintain a low vaginal pH, which is crucial for preventing harmful microbial overgrowth. The high numbers of lactobacilli, including LB931, had significantly lower vaginal pH levels, which is associated with a reduced prevalence of Group B streptococci (GBS) and yeast infections.

#### 3.3.7. ICU Patients

The potential of probiotics in critically ill patients, particularly ICU patients, has been explored. *L. plantarum* 299v [92] did not show significant improvements in clinical outcomes for ICU patients, suggesting that while probiotics may offer benefits in other settings, their role in critically ill patients requires further investigation. *L. plantarum* 299v is a well-studied probiotic strain recognized for its resilience in the gastrointestinal tract, surviving gastric acidity and bile exposure. It has demonstrated antimicrobial properties, reducing bacterial translocation and systemic inflammation, particularly in critically ill patients. Despite its ability to attenuate systemic inflammation and support gut microbiota restoration, the study found no significant results.

### 3.4. Central Nervous System

#### 3.4.1. Autistic Spectrum Disorders (ASD)

The specific probiotic strain *L. plantarum* PS128 [16,41] has shown significant improvement in symptoms associated with autism spectrum disorders (ASD). *L. plantarum* PS128 was also linked to improvements on the SNAP-IV scale, a widely used measure for ADHD and ASD symptoms, suggesting its potential adjunctive role in cognitive and behavioral therapy for ASD patients. *L. plantarum* PS128 has the ability to modulate gut microbiota and influence neurotransmitter levels, such as dopamine and serotonin, in the brain. Additionally, the probiotic was well-tolerated, with fewer side effects.

#### 3.4.2. Tourette Syndrome

*L. plantarum* PS128 [19] did not exhibit tic-reducing effects in children with Tourette syndrome. Tourette syndrome results from a complex interaction between social–environmental factors, multiple genetic abnormalities, and neurotransmitter disturbances. The present study reflected that intervention with PS128 did not have a superior response in tic severity improvement. There are a few possible explanations for this. *L. plantarum* PS128 has been demonstrated in germ-free mice to increase concentrations of dopamine, serotonin, and their metabolites in the striatum. However, it is known that patients with Tourette syndrome may have dopamine hypersensitivity and, therefore, respond to dopamine blocking agents such as risperidone, pimozide, and haloperidol. Therefore, it may be explainable that the use of PS128 did not show a superior response in improving tic severity. This suggests that while *L. plantarum* PS128 may be beneficial for ASD, its therapeutic benefits may not extend to all neurological or psychological conditions.

#### 3.4.3. Psychological and Cognitive Function

Several *L. plantarum* strains have been investigated for their impact on psychological well-being and cognitive function. *L. plantarum* HEAL9 [72] significantly reduced acute and chronic stress, indicating its stress-relieving potential. It showed a significant reduction in awakening cortisol levels, a key marker of stress response. Additionally, improvements were observed in mood, particularly in reducing confusion, anger, and depressive symptoms. The probiotic also enhanced cognitive function, particularly in memory and learning tasks, suggesting its role in modulating the gut–brain axis to alleviate stress and improve mental well-being.

*L. plantarum* P8 [88] has demonstrated broad benefits in stress management, including relief from stress, anxiety, and cognitive impairment in stressed adults, reinforcing its role in mental clarity enhancement. *L. plantarum* JYLP-326 [75] was found to alleviate anxiety, depression, and insomnia in college students experiencing test anxiety, highlighting its possible role in managing stress-induced psychological conditions. *L. plantarum* HEAL9 [73] was further linked to enhanced cognitive abilities in adults under moderate stress. *L. plantarum* DR7 [69] exhibited efficacy in reducing stress and anxiety in adults, supporting its role in mental well-being.

#### 3.4.4. Cognitive Health and Aging

In elderly populations, *L. plantarum* OLL2712 [74] demonstrated memory function decline-protective effects, suggesting its potential in maintaining cognitive health in aging individuals by modulating neuroinflammation through the microbiota–gut–brain axis. The probiotic’s high interleukin-10 (IL-10)-inducing activity helped to suppress neuroinflammation by inhibiting pro-inflammatory cytokines such as tumor necrosis factor-alpha (TNF-α) and interleukin-1β (IL-1β). Therefore, *L. planatrum* OLL2712’s immunomodulatory properties may help to mitigate age-related cognitive decline and neurodegenerative processes.

#### 3.4.5. Depression and Sleep Regulation

*L. plantarum* PS128 [68] showed notable reductions in depressive severity among patients with major depressive disorder (MDD), reinforcing its potential as a probiotic for depression management. *L. plantarum* PS128 showed significant decreases in the Hamilton Depression Rating Scale-17 (HAMD-17) and Depression and Somatic Symptoms Scale (DSSS) scores, indicating improved mood and reduced somatic symptoms. It also influence neurotransmitter pathways and gut permeability markers, such as zonulin and intestinal fatty acid-binding protein (I-FABP), suggesting a role in regulating systemic inflammation and neuroimmune interactions in MDD.

Additionally, *L. plantarum* PS128 [70] was found to reduce depressive symptoms and improve sleep quality in patients with insomnia, suggesting its broader role in mood elevation and sleep regulation. 

### 3.5. Integumentary System

#### 3.5.1. Skin Health

Several *L. plantarum* strains have demonstrated potential benefits in skin health, particularly in melanin regulation, anti-aging effects, and skin rejuvenation. *L. plantarum* GMNL6 [44] has shown melanin-reducing effects, suggesting its potential in addressing hyperpigmentation and age spots. *L. plantarum* GMNL6 significantly decreased melanin synthesis, potentially through the inhibition of the mitogen-activated protein kinase (MAPK) signaling pathway and the activation of protein kinase B (AKT), which suppresses melanogenesis. Additionally, lipoteichoic acid (LTA) from *L. plantarum* GMNL6 was identified as a key component contributing to its skin-lightening effects. A clinical trial further confirmed that a heat-killed *L. plantarum* GMNL6 cream significantly reduced melanin index (M index).

*L. plantarum* HY7714 [42] has been proven to reduce visible signs of aging, likely due to its antioxidant properties and ability to regenerate skin cells. *L. plantarum* LB224R ointment [43] has demonstrated anti-aging benefits, enhancing skin elasticity and reducing wrinkles, making it a promising candidate for preserving youthful skin appearance.

#### 3.5.2. Oral Health

*L. plantarum* 299v [56] was found to be as effective as chlorhexidine in reducing pathogenic bacteria in the oropharynx of chronically ill patients on mechanical ventilation, indicating its potential as an alternative strategy to prevent infections in vulnerable patients.

In pediatric oral health, *L. plantarum* CCFM8724 [57] was effective in preventing and treating early childhood caries, highlighting its role in dental health maintenance. It effectively prevented and treated early childhood caries (ECC) by modulating oral microbiota and reducing pathogens. *L. plantarum* CCFM8724 also significantly decreased *Streptococcus mutans* and *Candida albicans* while increasing beneficial genera like *Granulicatella* and *Gemella*. The probiotic also reduced caries-associated bacteria, helping to restore microbial balance and prevent biofilm formation, making it a promising ECC intervention.

### 3.6. Miscellaneous

#### 3.6.1. Lipid Profile

Several *L. plantarum* strains and probiotic combinations have demonstrated significant effects on lipid metabolism, particularly in regulating cholesterol and triglyceride levels. *L. plantarum* EGCG 13110402 [9], significantly reduced LDL, increased HDL, and helped to regulate blood pressure in individuals with normal to mildly hypercholesterolemic conditions.

*L. plantarum* Inducia [17] showed a significant reduction in total cholesterol, LDL-c, and non-HDL cholesterol, supporting its role in lipid metabolism.

The combination of *L. plantarum* strains CECT 7527, CECT 7528, and CECT 7529 [45,46] effectively lowered cholesterol levels in hypercholesterolemic patients, highlighting the potential of probiotic combinations in lipid management.

*L. plantarum* K50 [47] demonstrated potency in reducing total cholesterol and triglycerides, making it a promising probiotic for cardiovascular health improvement. *L. plantarum* K50 altered gut microbiota by increasing *Lactobacillus plantarum* and reducing *Actinobacteria*, changes correlated with visceral adiposity modulation. These findings suggest *L. plantarum* K50’s potential as a microbiome-targeted intervention for dyslipidemia management.

*L. plantarum* 299v [10] improved vascular endothelial function and showed potential in cardiovascular disease prevention. However, it did not significantly alter plasma cholesterol, fasting glucose, or body mass index.

#### 3.6.2. Blood Glucose Levels

*L. plantarum* HAC01 [11] significantly reduced 2 h postprandial glucose (2h-PPG) and HbA1c levels, suggesting its potential for managing blood glucose in individuals with prediabetes or type 2 diabetes. Despite no significant changes in fasting plasma glucose (FPG), the homeostasis model assessment of insulin resistance (HOMA-IR), and the quantitative insulin sensitivity check index (QUICKI), *L. plantarum* HAC01 may exert its effects via gut microbiota modulation rather than direct insulin sensitization.

#### 3.6.3. Anti-Obesity

The impact of *L. plantarum*’s role on weight control and obesity management has been explored in various studies. *L. plantarum* Dad-13 [79] did not result in significant changes in body weight or lipid profiles, suggesting strain-dependent effects. *L. plantarum* IMC 510 [69] demonstrated positive effects on weight control, suggesting its potential use in obesity management.

*L. plantarum* LMT-48 [80] was shown to induce weight loss, indicating its possible application in obesity prevention and treatment. The probiotic also reduced serum insulin levels, the homeostasis model assessment of insulin resistance (HOMA-IR), and leptin levels, indicating improved metabolic regulation. Additionally, gut microbiota analysis revealed increased *Lactobacillus* and *Oscillibacter* levels, with the latter being negatively correlated with triglyceride and alanine transaminase (ALT) levels, suggesting a role in lipid metabolism and adiposity control. These findings support *L. plantarum* LMT1-48’s therapeutic potential as a microbiome-targeted intervention for obesity management.

#### 3.6.4. Physiological Adaptations

Probiotics have also been investigated for their role in physical performance and post-exercise recovery. *L. plantarum* PS128 [14,15] significantly enhanced endurance performance in triathletes, particularly improving running performance and physiological adaptations related to athletic training. *L. plantarum* PS128 [77] was found to improve muscle recovery and renal function following a half marathon, highlighting its potential in post-exercise recovery and reducing exercise-induced damage.

*L. plantarum* TWK10 [78] demonstrated better physiological adaptation in non-athletic individuals, suggesting its broad application in enhancing physical performance and overall health. The probiotic also improved fatigue-associated biomarkers, reducing serum lactate and ammonia accumulation during exercise and recovery phases. Furthermore, *L. plantarum* TWK10 supplementation led to favorable body composition changes, including reduced fat mass and increased muscle mass. These effects may be linked to enhanced short-chain fatty acid (SCFA) production and gut microbiota modulation, suggesting *L. plantarum* TWK10 as a promising ergogenic aid.

## 4. Discussion

Overall, the findings show that *L. plantarum* strains have complicated and diverse functions for different physiological systems. Therapeutic efficacy is often different from one strain to another; thus, probiotic effects should be analyzed on a case-by-case basis with respect to the genetics of the microbes, interaction with hosts, and environmental influences. Such evidence indicates that some *L. plantarum* may greatly affect health outcomes because of its diverse modes of action; for instance, *L. plantarum* PS128 [16,68], confers protective effects against ASD, with beneficial effects in mitigating depression, stress, and sleep disturbances, probably by means of interactions with the gut–brain axis and neurotransmitter activity regulation. Another *L. plantarum* that may provide benefits to the vascular endothelial structure and microbial diversity, albeit not the best for IBS conditions, is *L. plantarum* 299v [10]. Inevitably, differences in the effectiveness of probiotics include contradictory results in many studies regarding the association with *L. plantarum* 299v [81]. This could be due to the heterogeneity of designs used in studies, the innate variation of the host response, and specific diseases as well. Those strains have a tendency to produce short-chain fatty acids and modulate inflammatory status; however, they often have insufficient strength against such complicated diseases as IBS. In such cases, effectiveness differs with IBS subtypes.

Among the lactic acid bacteria, *L. plantarum* 299v [13] turned out to be the most heterogeneous concerning their effects, which has been attributed to many factors, including differences in the design of studies. It has mainly to do with the difference in the approach taken by the different studies regarding the assessment of a range of symptoms. Also, the host is a factor determining the effect of *L. plantarum* 299v [92], as those with different gut microbiota and immune profiles respond differently. The effectiveness of *L. plantarum* 299v [38] may also depend on the subtype of IBS since its efficacy may be more pronounced for some subtypes than others. Just as *L. plantarum* MF1298 [59] fails to improve IBS symptoms, likely due to its poor colonization ability, *L. plantarum* P-8 enhances microbiome diversity through the reduction in opportunistic pathogens, underscoring that probiotic efficacy varies by strain.

Other factors that might influence probiotic efficacy include environmental and lifestyle elements such as diet, fiber intake, and prebiotics, which can shape the gut environment and affect the survival of a given probiotic. The ineffectiveness of *L. plantarum* CJLP243 [48] may be attributed to gut microbiome alterations following ileostomy in rectal cancer patients, whereas *L. plantarum* DR7 [69] may have alleviated stress and anxiety through gut neurotransmitter modulation, an effect influenced by dietary factors. The question remains open as to whether *L. plantarum* strains yield the best effects when administered alone or in combination with others. Some studies reported advantages for multi-strain formulations, as seen in the cholesterol-lowering effects of *L. plantarum* CECT 7527, CECT 7528, and CECT 7529 [46]. While other studies using the *L. plantarum* strain alone can demonstrate effective results as well.

Unfavorable or adverse effects from using *L. plantarum* are rare and not severe side effects. In one study [81], it was stated that the incidence of children with at least one adverse event was significantly higher in the placebo group compared with the *L. plantarum* 299V group. The most frequently recorded adverse events were pyrexia, headache, rash, anorexia, viral infection, and ear pain. There were no serious adverse events reported in the study.

## 5. Conclusions

Through highlighting the therapeutic versatility of *L. plantarum* strains across different physiological systems, it also observed how their inefficacies are expressed strain-specifically in the establishment of gastrointestinal health, immune modulation, central nervous system disorders, metabolic regulation, skin health, and physical performance. For example, irritable bowel syndrome, diarrhea, constipation, and bowel dysfunction symptoms have been observed to efficiently improve with strains such as *L. plantarum* Lpla33 and *L. plantarum* 299v, perhaps as a result of mechanisms involving gut barrier enhancement and inflammation modulation. Whereas *L. plantarum* IS-10506 increases mucosal immunity, *L. plantarum* PS128 seems to operate either via the gut–brain axis or closer peripheral sites to change certain neuropsychological functions. Other strains prove beneficial for improving metabolic health; for instance, *L. plantarum* EGCG 13110402 and *L. plantarum* HAC01 are implicated in downregulating lipid metabolism and reducing blood glucose levels. These findings empirically suggest that *L. plantarum* strains can be possible significant treatment adjuncts, particularly for applications in personalized medicine and integrative healthcare.

Regarding the limitations of our study, although this study tries to compile evidence from many sources, most findings are qualitative without effect size or confidence interval estimates. Study design variations, such as dosage, intervention period, and demographic characteristics in the endpoints above, also restrict external validity and render some findings not really applicable to the wider population. Confounding variables like diet, age, sex, genetic differences, and gut microbiome diversity, which can change the response to treatment, also have not been studied in this research. The potential confounding factors affect strain efficacy, such as creating differences in the host microbiome or dietary influences. Therefore, some *L. plantarum* strains shown inconsistent results, such as *L. plantarum* 299v, which shows mixed results; some studies show some positive and some negative effects. There may be several related factors, but confounding factors may play a major role in this issue.

Future research should employ randomized controlled trials (RCTs) with standardized dosages, treatment durations, adverse or side effects, and the validated therapeutic efficacy of *L. plantarum* strains, ensuring statistical significance with effect size and confidence intervals. Moreover, machine learning models could enhance personalized medicine by predicting patient-specific responses based on microbiome, genetic, and lifestyle factors. Furthermore, other promising *L. plantarum* strains in animal studies still require clinical trials to validate their role in precision medicine [18,75,93,94,95,96,97,98,99,100,101,102,103,104,105,106,107,108].

While this study applied a rigorous and systematic methodology to evaluate the clinical applications of *L. plantarum* strains, it is acknowledged that certain claims may still lack a critical evaluation due to inherent limitations in the included studies. One such limitation is the decision to restrict the literature search to Embase and PubMed, although this was justified by their strong biomedical relevance. Additionally, while stringent inclusion criteria were applied to ensure high-quality evidence, variations in study design, sample size, and clinical endpoints among the included studies could impact the strength of the conclusions drawn. To mitigate these limitations, a cautious and evidence-based interpretation of findings was adopted, highlighting the need for further well-controlled clinical trials to validate the therapeutic potential of *L. plantarum* strains.

In summary, most of the studies reviewed utilized only a single strain of *Lactobacillus plantarum* as a therapeutic agent to improve specific aspects of human health and well-being. However, the present study uniquely compiles multiple individual strains of *L. plantarum* and their diverse health applications for humans into one comprehensive overview. Their use in therapies must still be well substantiated with more rigorously controlled clinical studies. The periodic advances in research are expected to include microbiome profiling, precision probiotic interventions, and probiotic multi-strain synergies to maximize efficacy in therapy. More direct treatments may be considered for use in the future, such as Fecal Microbiota Transplantation (FMT), which could represent a promising advanced strategy to restore the gut microbiome and enhance the cognitive functions of healthy people and patients with neurological disorders. With the current and continuing expansion of probiotic science, *L. plantarum* will surely have the spotlight in the personalized, technologically enhanced, and evidence-based probiotic therapies that mainstream healthcare will use in the future.

## Figures and Tables

**Figure 1 nutrients-17-01165-f001:**
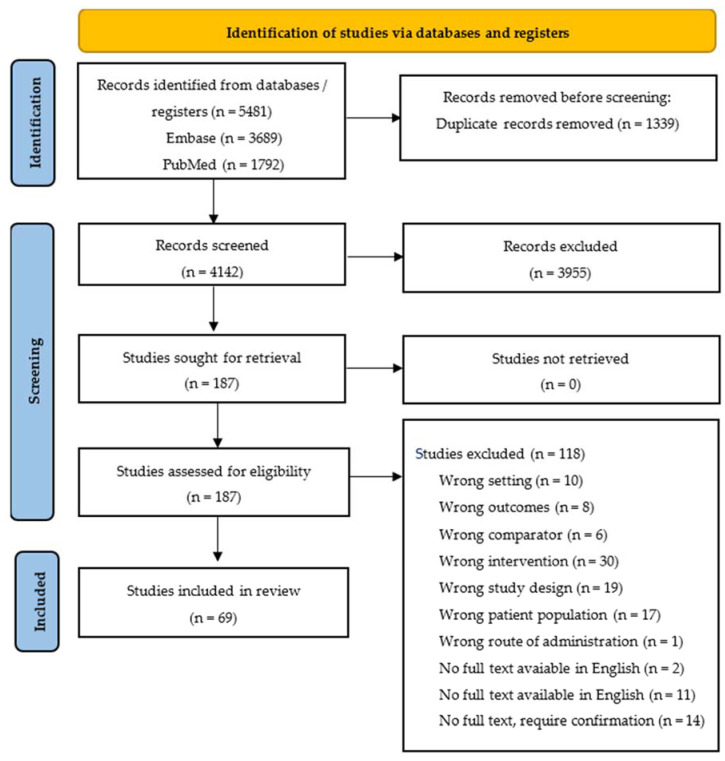
PRISMA 2020 flow diagram for new systematic reviews, which included the search of databases and registers only.

**Figure 2 nutrients-17-01165-f002:**
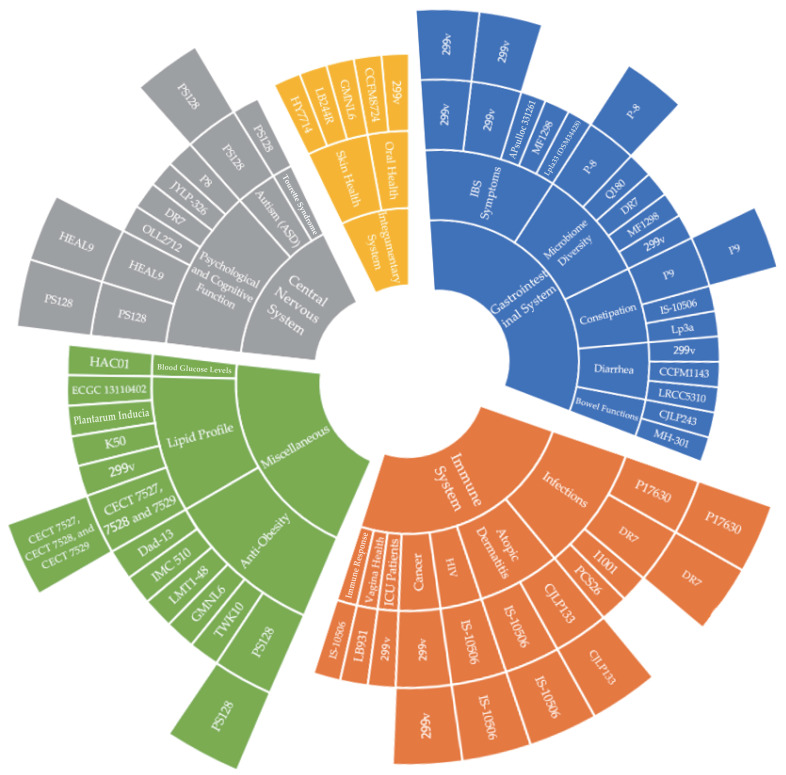
*Lactiplantibacillus plantarum* strains classification and distribution graph.

**Table 1 nutrients-17-01165-t001:** *Lactiplantibacillus plantarum’s* classification by symptoms with reference numbers.

Gastrointestinal System	Immune System	Central Nervous System	Integumentary System	Miscellaneous
IBS Symptoms [33,34,35,36,37,38,39]	Immune Response [40]	Autism (ASD) [16,41]	Skin Health [42,43,44]	Lipid Profile [9,10,17,45,46,47]
Bowel Functions [48,49]	Infections [50,51,52,53,54,55]	Tourette Syndrome [19]	Oral Health [56,57]	Blood Glucose Levels [11]
Microbiome Diversity [58,59,60,61,62,63]	Atopic Dermatitis [64,65,66,67]	Psychological and Cognitive Function [68,69,70,71,72,73,74,75]		Anti-Obesity [14,15,76,77,78,79,80]
Diarrhea [81,82,83]	Cancer [84,85]			
Constipation [86,87,88,89]	HIV [12,90]			
	Vagina Health [91]			
	ICU Patients [92]			

**Table 2 nutrients-17-01165-t002:** Table of results: A summarization of the *L. plantarum* strains’ therapeutic potential on various health conditions (from Appendix A).

*L. plantarum* Strain	Study Design	Health Condition	Key Clinical Findings
299v	Double-blind, placebo-controlled	Gastrointestinal symptoms, IBS, ICU, weight loss prevention	Mixed results: some studies positive, some negative effects
APsulloc 331261 (GTB1(TM))	A randomized, double-blind, and placebo-controlled trial	Intestinal discomfort symptom	Enhanced intestinal discomfort symptoms, defecation consistency, quality of life, beneficial microbiota, and overall intestinal health
CCFM1143	Randomized, double-blind, placebo-controlled trial	Chronic diarrhea	Probiotics showed clinical effectiveness in managing chronic diarrhea
CCFM8724	Randomized, double-blind, placebo-controlled	Early childhood caries (ECC)	Reduced oral pathogens
CECT 7527, 7528 and 7529	A controlled, randomized, double-blind trial.	Hypercholesterolemic patients	Significant reduction in serum cholesterol in hypercholesterolemic patients.
CJLP133	N.A.	Atopic dermatitis (AD)	Improvement of atopic dermatitis (AD)
CJLP243 (KCCM11045P)	Randomized, double-blind, placebo-controlled	Bowel function	Improved bowel function and quality of life
DR7	Randomized, double-blind, placebo-controlled	Stress, anxiety, cognitive function, URTI	Reduced anxiety, improved cognition, reduced URTI symptoms
Dad-13	A randomized, double-blind, placebo-controlled study	Body weight, BMI	A significant decrease in body weight and BMI (*p* < 0.05). Dad-13 also caused the Firmicutes population to decrease and the Bacteroidetes population to increase these
ECGC 13110402	Randomized, double-blind, placebo-controlled	Hypercholesterolemia	Cholesterol-lowering effect
Lpla33 (DSM34428)	Randomized, double-blind, placebo-controlled	IBS symptom	The Lpla33 (DSM34428) was well tolerated and helped to improve irritable bowel syndrome (IBS)
GMNL6	Randomized, double-blind, placebo-controlled	Skin condition	Improved skin conditions
HAC01	A randomized, double-blind, placebo-controlled clinical trial was conducted	HbA1c and 2h-PPG levels	Significantly improved HbA1c and 2h-PPG levels
HEAL9	A randomized, double blind, placebo-controlled study	Inflammatory markers, cognitive functions	Reduced inflammatory markers and cognitive functions
HY7714	Randomized, double-blind, placebo-controlled	Skin elasticity, anti-aging	Improved skin elasticity
IMC 510	Randomized, placebo-controlled	Obesity and metabolic health	Reduced body weight and improved metabolic parameters
I1001	Non-randomized parallel cohorts	Vulvovaginal candidiasis (VVC)	Prevented recurrence of vulvovaginal candidiasis (VVC)
Inducia	Randomized, double-blind, placebo-controlled	Metabolic syndrome, cholesterol reduction	Improved cholesterol levels and metabolic outcomes
IS-10506	Randomized, double-blind, placebo-controlled	Atopic dermatitis, immune modulation, constipation	Immunomodulatory effects, improvement in dermatitis and SCFA modulation
JYLP-326	Randomized, placebo-controlled	Anxiety, depression, insomnia	Reduced anxiety and depression symptoms
K50	Randomized, double-blind, placebo-controlled	Hyperlipidemia	Reduced cholesterol and triglycerides
LB244R®	Single-center, topical ointment study	Skin anti-aging	Improved anti-aging effects
LB931	Randomized, placebo-controlled	Vaginal health, Group B streptococci reduction	Reduced vaginal pH, decreased pathogenic colonization
LMT1-48	A randomized, double-blind, placebo-controlled clinical trial	Body weight	Decreased body weight
Lp3a	Randomized, placebo-controlled, double-blind	Functional constipation	Improved constipation
LRCC5310	Randomized, placebo-controlled	Rotaviral gastroenteritis	Improved gastroenteritis symptoms, inhibited viral proliferation
MF1298	Randomized, double-blind, placebo-controlled, crossover	Gut microbiota and symptoms	Directly effected gut health
MH-301	A single-center, randomized, double-blind, and placebo-controlled clinical trial	Bowel movements	Improved bowel movements
OLL2712	A randomized, double-blind, placebo-controlled trial	Memory function decline in older adults	Protective effects against memory function decline in older adults
P17630	Randomized, double-blind, placebo-controlled	Vulvovaginal candidiasis	Prevention of VVC recurrence
P-8	Randomized, placebo-controlled	Gut microbiota modulation	Beneficial gut microbiota modulation
P8	A randomized, double-blind and placebo-controlled study, N.A.	Stress, anxiety, memory and cognitive symptoms in stressed adults, fecal bacterial structure, fecal microbiota, SIgA, SCFAs, and TBAs	Alleviation of selected stress, anxiety, memory, and cognitive symptoms in stressed adults. Modulated fecal microbiota, SIgA, SCFAs, and TBAs in healthy individuals
P9	Randomized, double-blind, placebo-controlled	Chronic constipation	Improved constipation symptoms
PC26	A double-blind, randomized, placebo-controlled clinical pilot study	Urinary tract infections (UTIs)	Helpful in alleviating urinary tract infection (UTI) symptoms and UTI prevention
PS128	Randomized, double-blind, placebo-controlled	Anxiety, depression, autism, ADHD	Improved mental health and cognitive function, reduced anxiety symptoms
Q180	A double-blind, randomized, placebo-controlled study	Postprandial lipid levels	Helped to maintain healthy postprandial lipid levels through modulating gut environment
TWK10	A double-blind, placebo-controlled experiment	Body fat	Significantly decreased body fat

## Data Availability

The data used and analyzed in this systematic scoping review were obtained from publicly available sources, including the PubMed and Embase databases. All relevant data supporting the findings of this study are included in the manuscript. No additional raw data were generated or used beyond those publicly accessible.

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
