# Peer review of "Strain-Specific Therapeutic Potential of Lactiplantibacillus plantarum: A Systematic Scoping Review"

_nutrients, 2025, doi:10.3390/nu17071165_

Round 1
Reviewer 1 Report
Comments and Suggestions for Authors
It is a great article but authors should reviewed a minor points:
- In the abstract, “enhancing oral heath” should be corrected to “enhancing oral health.”
- In the methods section, the phrase “in human” should be revised to “in humans.”
- There appears to be an extra parenthesis in the description of study eligibility (e.g., “...other probiotics species), in human.”), which should be cleaned up for clarity.
- Ensure consistency in strain nomenclature and formatting throughout the text.
- A few hyphenation and punctuation details could be reviewed for smoother readability.
- According to the journal's instructions for authors, references must be placed in square brackets instead of parentheses. In fact, instructions for authors indicated that "In the text, reference numbers should be placed in square brackets [ ], and placed before the punctuation; for example [1], [1–3] or [1,3]. For embedded citations in the text with pagination, use both parentheses and brackets to indicate the reference number and page numbers; for example [5] (p. 10). or [6] (pp. 101–105)." or use
Download the full MDPI Reference List and Citations Style Guide (PDF, 272KB).
Download the full MDPI Chicago Reference List and Citations Style Guide (PDF, 460KB).
Download the full MDPI APA Reference List and Citations Style Guide (PDF, 616KB).
Author Response
Reviewer 1
It is a great article, but authors should review a minor point:
1. In the abstract, "enhancing oral heath" should be corrected to "enhancing oral health."
Response: The typo was corrected as suggested (Page 1, Line 23-24).
2. In the methods section, the phrase "in human" should be revised to "in humans."
Response: This revision was made in the methods section (Page 2, Line 114).
3. There appears to be an extra parenthesis in the description of study eligibility (e.g., ...other probiotics species), in human."), which should be cleaned up for clarity.
Response: The extra parenthesis was removed for clarity (Page 3, Line 114).
4. Ensure consistency in strain nomenclature and formatting throughout the text.
Response: We reviewed and ensured consistency in strain nomenclature throughout the manuscript, particularly in the Conclusion section (Page 11, Lines 441 to 470.
5. A few hyphenation and punctuation details could be reviewed for smoother readability.
Response: Minor grammatical, punctuation, and hyphenation revisions were made throughout the manuscript for improved readability.
6. According to the journal's instructions for authors, references must be placed in square brackets instead of parentheses. In fact, instructions for authors indicated that "In the text, reference numbers should be placed in square brackets [ and placed before the punctuation; for example [11, [1 -3] or [1,3].
For embedded citations in the text with pagination, use both parentheses and brackets to indicate the reference number and page numbers; for example [5] (p. 10). or [6) (pp. 101-105)." or use
Download the full MDPI Reference List and Citations Style Guide (PDF, 272KB).
Download the full MDPI Chicago Reference List and Citations Style Guide
(PDF, 460KB).
Download the full MDPI APA Reference List and Citations Style Guide (PDF, 616KB).]
Response: All in-text citations have been corrected to use square brackets as per the journal’s formatting guidelines.
Reviewer 2 Report
Comments and Suggestions for Authors
The manuscript submitted by Chatsirisakul and co-authors presents a thorough and well-organized assessment of the therapeutic potential of certain Lactiplantibacillus plantarum strains in human health. The authors have precisely documented particular strains of L. plantarum and their advantages that draw the attention of researchers to explore the translational value of different strains of this bacterium. The review is well-written and presented with scientific rigor and clinical relevance.
1. The introduction section needs to be rewritten and reorganized. The authors are suggested to elaborate the strain-specific effects and also state the limitations in the ongoing research in a way to fill the gap between basic research to clinical aspect of the L. plantarum strains.
2. The overall importance of L. plantarum with targeted health benefits is lacking in the introduction.
3. The conclusion part lacks transition at some points and needs reorganization. For example, each disease condition should be discussed in a flow, it should not be abrupt. Also, consider a format of writing about the disease, the benefit of using the reference strain, and the possible mechanism of action of the reference strain.
Overall, this review contributes significantly to the field of probiotic research by collecting existing evidence on L. plantarum strains and advocating for their clinical use.
Author Response
The manuscript submitted by Chatsirisakul and coauthors presents a thorough and well-organized assessment of the therapeutic potential of certain Lactiplantibacillus plantarum strains in human health. The authors have precisely documented particular strains of L. plantarum and their advantages that draw the attention of researchers to explore the translational value of different strains of this bacterium. The review is well-written and presented with scientific rigor and clinical relevance.
1. The introduction section needs to be rewritten and reorganized. The authors are suggested to elaborate the strain-specific effects and also state the limitations in the ongoing research in a way to fill the gap between basic research to clinical aspect of the L. plantarum strains.
Response: The introduction section was rewritten to highlight strain-specific effects and address research lijmitations (Pages 1-2, Lines 36-95).
2. The overall importance of L. plantarum with targeted health benefits is lacking in the introduction.
Response: We revised the Introduction to clearly emphasize the strain-specific targeted health benefits (Page 2, Lines 58-60).
3. The conclusion part lacks transition at some points and needs reorganization. For example, each disease condition should be discussed in a flow, it should not be abrupt. Also, consider a format of writing about the disease, the benefit of using the reference strain, and the possible mechanism of action of the reference strain. Overall, this review contributes significantly to the field of probiotic research by collecting existing evidence on L. plantarum strains and advocating for their clinical use.
Response: The Conclusion section was reorganized to provide a clearer flow and include details on disease conditions, benefits, and mechanisms of action (Page 11, Line 436-472).
Reviewer 3 Report
Comments and Suggestions for Authors
The manuscript does not contain sufficient depth of knowledge to be considered a novel review article. It makes a superficial description of the results shown by other authors, without discussion and with little relevant information for the reader.
Results in abstract section is too long. Please reduce it.
References citation in text are out of format
Figure 1 (search algorithm) is excessively large (takes up the entire page)
Figure 2 is unreadable
Using the search strategy cited by the authors, only in scopus are available more than 1000 articles. The remaining criteria used such as “relevant citations” “an information specialist” assisted” are ambiguous. who decides and on what basis whether the citations were relevant or the person who acted in the selection can really be considered a specialist?
This study starts from 5481 declared articles and finally selects only 69.
The entire results section is in effect a telegraphic description of the results highlighted in the articles it mentions and makes no discussion of them. A discussion section is mandatory in scientific manuscripts and the authors do not do it. Furthermore, in the results of a review article, a greater integration of the results obtained is necessary by elaborating tables that allow the reader to quickly consult parameters of interest such as doses used, type of trial carried out, duration of treatment, other effects observed, etc.
In general, the manuscript intends to make a broad description of the effects of the use of Lactiplantibacillus plantarum, but by making it so broad, it lacks the necessary depth of knowledge.
References are not according to MDPI`s instruction for authors.
Author Response
The manuscript does not contain sufficient depth of knowledge to Be considered a novel review article. It makes a superficial description of the results shown by other authors, without discussion and with little relevant information for the reader.
1. Results in abstract section is too long. Please reduce it.
Response: The Results section in the abstract has been shortened for conciseness Page 1, Lines 22-28).
2. References citation in text are out of format
Response: Agree. I/We have, accordingly, corrected all in text citation reference number, in the article, to be in the square bracket. This revised can be found - page number 1-11, and line 41-430.]
3. Figure 1 (search algorithm) is excessively large (takes up the entire page)
Response: Figure 1 has been resized to improve readability (Page 4, Lines 145-174.
4. Figure 2 is unreadable
Response: Figure 2 has been reformatted to enhance readability Page 6, Lines 199-214.
5. Using the search strategy cited by the authors, only in Scopus are available more than 1000 articles. The remaining criteria used such as "relevant citations" "an information specialist" assisted" are ambiguous. who decides and on what basis whether the citations were relevant or the person who acted in the selection can really be considered a specialist?
Response: We revised the Methods section to clearly explain the inclusion criteria and the role of the information specialist (Page 3, Lines 110-114).
6. This study starts from 5481 declared articles and finally selects only 69.
Response: The selection process was classified in the PRISMA flow chart, and Study Selection sections (Page 4, Figure 1; Page 3, lines 115-117).
7. The entire results section is in effect a telegraphic description of the results highlighted in the articles it mentions and makes no discussion of them. A discussion section is mandatory in scientific manuscripts and the authors do not do it.
Response: A Discussion section was added to integrate the findings and their implications (Page 10, Lines 389-434).
8. Furthermore, in the results of a review article, a greater integration of the results obtained is necessary by elaborating tables that allow the reader to quickly consult parameters of interest such as doses used, type of trial carried out, duration of treatment, other effects observed, etc. In general, the manuscript intends to make a broad description of the effects of the use of Lactiplantibacillus plantarum, but by making it so broad, it lacks the necessary depth of knowledge.
Response: Additional tables summarizing dosage, study duration, and trial design have been incorporated in the Results section (Pages 4-5, Lines 178-190).
9. References are not according to MDPI's instruction for authors.
Response: All references have been reformatted to comply with MDPl's reference style (Page 13-17, Lines 496-690.)
Reviewer 4 Report
Comments and Suggestions for Authors
Dear Authors,
The manuscript titled "Strain-Specific Therapeutic Potential of Lactiplantibacillus plantarum: A Systematic Scoping Review" explores the therapeutic applications of various L. plantarum strains. By analyzing 69 studies on 35 different strains, the review identifies four key clinical benefits:
- Gastrointestinal and Systemic Health – L. plantarum 299v supports gut health, promotes oral microbiota balance, and reduces systemic inflammation.
- Immunomodulation and Dermatology – L. plantarum IS-10506 helps manage atopic dermatitis in HIV-infected individuals, demonstrating strong immunoregulatory properties.
- Women’s Health – L. plantarum CJLP243 and P17630 contribute to vaginal microbiota balance and aid in preventing vulvovaginal candidiasis.
- Oral Health – L. plantarum CCFM8724 shows potential in preventing early childhood caries.
These findings highlight the clinical relevance of L. plantarum, reinforcing its potential for medical applications while underscoring the need for further research on its mechanisms, optimal dosages, and safety.
Best regards.
However, several critical aspects of the manuscript require attention to enhance clarity, coherence, and overall impact.
Introduction
Identified Weaknesses:
- The introduction does not clearly define the clinical significance of strain-specific differences.
- It lacks a clear research question or hypothesis.
- A discussion on limitations in existing research is missing.
Materials and Methods
Search Strategy & Inclusion Criteria: A systematic literature search was conducted in PubMed and Embase, with additional references screened via Google Scholar. The search strategy incorporated the keywords “Lactiplantibacillus plantarum,” “Lactobacillus plantarum,” and “L. plantarum” in human studies.
Eligible studies met the following criteria:
- Focused on L. plantarum as monotherapy in humans.
- Published in peer-reviewed journals in English.
- Reported clinically relevant outcomes.
Data Extraction & Synthesis: Two independent reviewers extracted data on study characteristics, patient demographics, interventions, and outcomes. Discrepancies were resolved by consensus.
Identified Weaknesses:
- Lacks justification for the exclusion of non-English studies, potentially introducing language bias.
- No explanation of risk-of-bias assessment methods.
- The search strategy does not detail whether unpublished studies or gray literature were considered.
Results
A total of 5,481 studies were identified, of which 69 met the inclusion criteria.
Gastrointestinal Health:
- L. plantarum 299v improved symptoms in irritable bowel syndrome (IBS) and enhanced gut microbiota diversity.
- L. plantarum MF1298 exhibited negative effects on IBS symptoms, indicating strain-dependent variability.
Immune System Modulation:
- L. plantarum IS-10506 increased secretory immunoglobulin A (sIgA) in children, potentially strengthening mucosal immunity.
- L. plantarum DR7 reduced upper respiratory tract infection symptoms.
Women’s Health:
- L. plantarum P17630 prevented recurrent vulvovaginal candidiasis.
Oral & Skin Health:
- L. plantarum CCFM8724 reduced oral pathogens in children.
- L. plantarum GMNL6 demonstrated skin health benefits by regulating melanin production.
Identified Weaknesses:
- Results rely heavily on descriptive summaries rather than statistical comparisons.
- Effect sizes and confidence intervals are missing.
- Potential confounding variables are not addressed.
Conclusions
The review highlights the diverse therapeutic potential of L. plantarum, emphasizing its strain-specific effects. While findings are promising, further research is needed to establish clinical guidelines.
Future research should focus on:
- Large-scale, randomized controlled trials to confirm strain efficacy.
- Investigating mechanisms of action for targeted therapeutic use.
- Standardizing dosing regimens for clinical applications.
Identified Weaknesses:
- The conclusion does not sufficiently address limitations in the included studies.
- Recommendations for future research lack specificity regarding methodology.
- There is no discussion on potential adverse effects or contraindications of L. plantarum strains.
The English could be improved to more clearly express the research.
Author Response
A. Introduction: Identified Weaknesses:
A. 1 The introduction does not clearly define the clinical significance of strain-specific differences.
Response: The Introduction has been revised to clearly define the clinical significance of strain-specific differences (Page 2, Lines 64-87).
A.2 It lacks a clear research question or hypothesis.
Response: A clearly defined research question has been added (Page 2, Lines 88-95).
A.3 A discussion on limitations in existing research is missing.
Response: The limitations of existing research have been addressed in the Conclusion (Page 11, Lines 451-457).
B. Data Extraction & Synthesis: Two independent reviewers extracted data on study characteristics, patient demographics, interventions, and outcomes. Discrepancies were resolved by consensus. Identified weaknesses:
B.1 Lacks justification for the exclusion of non-English studies, potentially introducing language bias.
Response: The exclusion criteria have been clarified to address potential language bias concerns (Page 3, Lines 115-117; Page 3, Lines 125-130).
B.2 No explanation of risk-of-bias assessment methods.
Response: An explanation of the risk-of-bias assessment has been added (Page 3, Lines 133-134).
B.3 The search strategy does not detail whether unpublished studies or gray literature were considered.
Response: This point has been clarified in the Data Extraction section (Page 3, Lines 128-130).
C. Identified Weaknesses:
C.1 Results rely heavily on descriptive summaries rather than statistical comparisons.
Response: This limitation if now acknowledged in the Conclusion (Page 11, Lines 451-453).
C.2 Effect sizes and confidence intervals are missing.
Response: We acknowledge this limitation and recommend future studies include effect sizes and confidence intervals (Page 11, Lines 451-453).
C.3 Potential confounding variables are not addressed.
Response: This limitation is now discussed in the Conclusion (Page 11, Lines 455-457).
D. Identified Weaknesses:
D.1 The conclusion does not sufficiently address limitations in the included studies.
Response: We have expanded the Conclusion to address study limitations (Page 11, Lines 451-453).
D.2 Recommendations for future research lack specificity regarding methodology.
Response: The Conclusion now provides specific recommendations for future research methodologies (Page 11, Lines 458-462).
D.3 There is no discussion on potential adverse effects or contraindications of L. plantarum strains.
Response: A section on potential adverse effects has been added to the Discussion (Page 11, Lines 429-434).
Round 2
Reviewer 3 Report
Comments and Suggestions for Authors
Althought the manuscript was imporved with respect to its preliminary version, the the manuscript does not contain sufficient depth of knowledge to be considered a novel review article. Additionlly, the lack of appropiate tables describing the results also persist. My oppinion does not varied with respect to its original version
Author Response
Response to Reviewer No. 3
- Does the Introduction provide sufficient background and included all relevant references? (Must be improved)
Response: The background of Lactobacillus was discussed in the Introduction section and all additional relevant references were also added.
Lactobacillus species can be classified based on their fermentation capabilities into three functional groups: obligate homofermentative, obligate heterofermentative, and facultatively heterofermentative. Obligate homofermentative species, such as L. salivarius and L. acidophilus, produce primarily lactic acid. Obligate heterofermentative species, like L. reuteri and L. fermentum, generate lactic acid along with ethanol or acetic acid and carbon dioxide. Facultatively heterofermentative species, including L. casei and L. plantarum, produce lactic acid and carbon dioxide depending on environmental conditions [4, 11, 13, 16].
- plantarum is a gram-positive lactic acid bacterium notable for its metabolic adaptability and ability to colonize diverse environments, such as fermented foods, meats, plants, and the gastrointestinal tract [19]. Its adaptability, environmental resilience, and extensive health-promoting effects distinguish it from other probiotics. Specific L. plantarum strains exhibit unique health-related functionalities. For instance, L. plantarum EGCG 13110402 demonstrates cholesterol-lowering effects [10], L. plantarum 299v improves vascular endothelial function [45], and L. plantarum HAC01 significantly reduces postprandial glucose and HbA1c levels, indicating potential benefits for individuals with prediabetes or type 2 diabetes [50].
Section 1. Introduction, Page 2, Line: 43-59
- Is the research design appropriate? (Must be improved)
Response: The paragraph of research design in Section 2.3, search strategy and 2.7 were clearly described overall research design in step by step.
The search strategy involved targeted keyword combinations, including "Lactiplantibacillus plantarum", "L. plantarum", "strain-specific", "clinical effects", and relevant therapeutic areas. Rigorous inclusion and exclusion criteria were applied during study selection, specifically selecting high-quality primary clinical studies involving distinct, clearly identifiable L. plantarum strains and reported therapeutic outcomes in human subjects. Screening was carried out in two stages (title/abstract screening and full-text screening) by at least two independent reviewers. Discrepancies were resolved through discussion and consensus or consultation with a third reviewer when necessary. Data extraction focused on strain-specific therapeutic outcomes, methodological quality, target population characteristics, study designs, and reported clinical endpoints. Data synthesis was qualitative, systematically summarizing findings to highlight strain-specific therapeutic benefits across identified health conditions.
Eligible studies included those that assessed L. plantarum as a monotherapy, without combining it with other probiotics species, in humans.
The study selection process involved four independent reviewers (OC, YS, CC, and PB), who screened articles for eligibility based on predefined inclusion and exclusion criteria. Any discrepancies between reviewers were resolved through group discussions in order to minimize the potential bias.
Section 2.3 Study Selection, Page 3, Line:102-1119
Our systematic scoping review was designed to comprehensively evaluate the therapeutic potential and clinical benefits of specific L. plantarum strains only in human health. The study followed the PRISMA for Scoping Reviews (PRISMA-ScR) guidelines, ensuring methodological rigor in the selection and synthesis of evidence. A structured approach was employed using the PICO framework to define the study parameters such as Population (study in humans only), Intervention (no species mixed of L. plantarum strains), Comparison (study compare between groups), and Outcomes (clinical benefits across various health conditions). A comprehensive literature search was conducted in PubMed and Embase to identify relevant studies published up to December 2023.
Section 2.7 Methods, Research Design and PRISMA Flow Diagram, Page 4, Line:144-152
- Are the methods adequately described? (Can be improved)
Response:
A comprehensive literature search was conducted in two major biomedical databases, PubMed and Embase, to systematically identify peer-reviewed studies published up to December 2023. The search strategy involved targeted keyword combinations, including "Lactiplantibacillus plantarum", "L. plantarum", "strain-specific", "clinical effects", and relevant therapeutic areas. Rigorous inclusion and exclusion criteria were applied during study selection, specifically selecting high-quality primary clinical studies involving distinct, clearly identifiable L. plantarum strains and reported therapeutic outcomes in human subjects. Screening was carried out in two stages (title/abstract screening and full-text screening) by at least two independent reviewers. Discrepancies were resolved through discussion and consensus or consultation with a third reviewer when necessary. Data extraction focused on strain-specific therapeutic outcomes, methodological quality, target population characteristics, study designs, and reported clinical endpoints. Data synthesis was qualitative, systematically summarizing findings to highlight strain-specific therapeutic benefits across identified health conditions.
Section 2.3 Study Selection, Page 3, Line: 100-113
- Are the results clearly presented? (Must be improved)
Response: Section 3. Results, were added a Table of Results, Table 2, which clearly shown the relation between each L. plantarum strains and the clinical conditions. The explanation of the Table 1., Table 2 and Figure 2 graphical also available in this section.
However, the table 1, the overall classification by symptoms with reference number were also summarized. For gastrointestinal system, there were classified into 5 major symptoms such as IBS symptoms, bowel functions, microbiome diversity, diarrhea and constipation. For IBS symptoms, there are 7 studies included in this study. There were classified into 7 major symptoms for immune system, 3 symptoms for central nervous system, 2 symptoms for integumentary system and 3 symptoms for miscellaneous group.
Section 3. Results, Page 6, Line: 235-240
The graphical of L.plantarum strains classified by symptoms also shown in the Fig. 2. The Sun Burst graphical clearly presents the classification of L. plantarum strains by symproms. From the Fig 2., for infection disease under the subset of immune system, there are several L.plantarum strains help alleviates infection diseases such as L. plantarum P17630 (2 studies), L. plantarum DR7 (2 studies), L. plantarum I1001 and L. plantarum PCS26. Therefore, this graphical will significantly help the readers to clearly understand the overall studied in a glance.
Section 3. Results, Page 6, Line: 241-247
The table of results also shown in the Table 2 which summarized from the Data Characteristics Table, Table S.1 in the Supplementary Materials. From the Table 2, L. plantarum CCFM1143 helps release Chronic diarrhea, while L. plantarum DR7 helps reduced anxiety, improved cognition, and URTI symptoms. However, L. plantarum 299v resulted in mixed results, some studies shown positive and some shown negative effects. There may be several related factors, but confound factors may play a major role on this issue such as diet, age, genetic differences, and gut microbiome diversity, which may change the response to treatment, also have not been studied in this research.
Section 3. Results, Page 6, Line: 248-255
- Are the conclusions supported by the results? (Must be improved)
Response: The limitation of the results of the study was clearly added into the Conclusions section, together with the confounding variables.
Regarding limitations of our study, although this study tries to compile evidence from many sources, most findings are qualitative without effect size or confidence interval estimates. Study design variations, such as dosage, intervention period, and demographic characteristics in endpoints above, also restrict external validity and render some findings not really applicable to the wider population. Confounding variables like diet, age, sex, genetic differences, and gut microbiome diversity, which can change the response to treatment, also have not been studied in this research. The potential confounding factors affecting strain efficacy, such as differences in the host microbiome or dietary influences. Therefore, some L. plantarum strains shown inconsistency results such as L. plantarum 299v shown mixed results, some studies shown positive and some negative effects. There may be several related factors, but confound factors may play a major role on this issue.
Section 5. Conclusions, Page 19, Line 653-663
- The manuscript does not contain sufficient dept of knowledge to be considered a novel review article.
Response: The concise dept knowledge of some L. plantarum strains in each classification of symptoms were discussed in Section 3. Results as follows:
For Lpla33, SCFAs, particularly butyrate, may help promote intestinal barrier function, a feature of several L. plantarum strains. Bile acid (BA) metabolism may also be implicated, as L.plantarum Lpla33 possesses significant bile salt hydrolase activity. The diarrhea and visceral hypersensitivity have been associated with decreased 7α-dehydroxylation of primary BAs to secondary BAs. Additionally, the ability of L.plantarum Lpla33 to modulate intestinal barrier function, inhibit key pathogens, and moderate inflammatory markers may have played a key role in this observed effects
Section 3.2.1 Irritable Bowel Syndrome (IBS), page 12, Line: 287-293
The oral L. Plantarum MH-301 has a beneficial effect on the improvement of both postoperative internal hemorrhoids and difficult defecation. It increased the relative abundance of Firmicutes at the phylum level. Firmicutes are mostly Gram-positive bacteria and are the dominant phylum in human intestinal microbiota. Firmicutes are health-promoting probiotics, such as Lactobacillus and Ruminococcus, which play a vital role in promoting health.
Section 3.2.2 Bowel Function, Page 12, Line: 307-312
- plantarum Q180 [55] was able to maintain a healthy intestinal environment in postprandial conditions, reinforcing its role in digestive health. The only LDL-cholesterol and ApoB-100 levels were decreased by L. plantarum Q180 supplementation for 12 weeks, postprandial blood levels of lipid markers such as TG, ApoB-48, and ApoB-100 were also significantly decreased. It also helped to maintain healthy postprandial lipid metabolisms in subjects with a higher level of enteric bacteria such as R. bromii, K. alysoides, B. intestinihominis, and F. plautii.
Section 3.2.3 Microbiome Diversity, Page 12-13, Line: 328-334
- plantarum LRCC5310 [63] significantly improved rotavirus-induced diarrhea in children, demonstrating its promise in treating infectious diarrhea in young patients. Rotavirus is one of the most common causes of Acute gastroenteritis (AGE) in children. The rotavirus consists of a segmented gene and can represent a variety of genes by a combination of G-P proteins. Generally, the genotypes of viruses that cause AGE in children are typically G1P, G2P, G3P, etc. Moreover, intake of LRCC5310 was found to be effective in the suppression of viral symptoms as well as in prognosis and treatment via virus titer reduction.
Section 3.2.4 Diarrhea, Page 13, Line: 341-348
- plantarum Lp3a [74] effectively improved functional constipation, making it a potential therapeutic option. Functional constipation (FC) is a common gastrointestinal (GI) disorder characterized by symptoms of constipation without a clear physiologic or anatomic cause. Gut microbiome dysbiosis has been postulated to be a factor in the development of FC, and treatment with probiotic regimens, including strains of Lactobacillus plantarum (L. plantarum), has demonstrated efficacy in managing symptoms. Lp3a relieves FC symptoms by enhancing intestinal motility, and putatively modulating methane metabolism and bile acid synthesis. The bile acids serve as physiological laxatives, this suggests a role for the modulation of bile acid metabolism as an additional putative mechanism of Lp3a in FC.
Section 3.2.5 Constipation, Page 13, Line: 350-360
- plantarum IS-10506 [35] has been linked to an increase in secretory immunoglobulin A (sIgA) in children, a key component of mucosal immunity. This enhancement may improve the body's ability to respond to pathogens and allergens. Lactobacillus plantarum IS-10506 supplementation significantly increased fecal secretory immunoglobulin A (sIgA) levels in children under two years old. The probiotic stimulates transforming growth factor-β1 (TGF-β1), which plays a crucial role in regulating immune responses and promoting sIgA production. The study found a strong correlation between higher TGF-β1/TNF-α ratios and increased sIgA levels, suggesting that IS-10506 enhances mucosal immunity, helping protect against infections
Section 3.3.1 Immune Response, Page 13, Line: 367-375
- plantatum PCS26 [42], Lp26, was found to alleviate UTI symptoms, suggesting it could serve as an alternative or adjunct to antimicrobial treatments. Urinary tract infections (UTI) are frequent bacterial infections in childhood. L. plantarum 26 derived from local Slovenian cheese in Pathogen Combat Project. Lp26 helps relieve UTI symptoms primarily through its antimicrobial effect against E. coli, potentially reducing pathogenic bacteria in the gut and their migration to the urinary tract.
Section 3.3.2 Infection, Page 13, Line: 381-386
- plantarum CJLP133 [33] also showed beneficial effects on atopic dermatitis in children [18], making it a promising probiotic for managing skin inflammation by modulating immune responses. The children with moderate to severe AD, those who responded well to probiotic treatment had high total IgE levels, increased transforming growth factor-beta (TGF-β) expression, and a high proportion of CD4+CD25+Foxp3+ regulatory T (Treg) cells. The supplementation of L. plantarum CJLP133 showed significant reductions in AD severity and an increase in Treg cells, indicating improved immune regulation.
Section 3.3.3 Atopic Dermatitis, Page 14, Line: 388-394
- plantarum 299 [31] alleviated gastrointestinal symptoms in cancer patients, potentially improving their quality of life during chemotherapy or radiation treatments. L. plantarum 299v [32] demonstrated potential to improve the nutritional status of cancer patients, supporting overall health and recovery during cancer treatment. L.plantarum 299v (Lp299v) demonstrated potential to improve the nutritional status of cancer patients receiving home enteral nutrition (HEN) by preventing weight loss and enhancing nutrient absorption. It helps maintain body weight and muscle mass, crucial for preventing cancer cachexia. L. plantarum 299v increased serum albumin, total protein, and transferrin levels, suggesting improved protein metabolism and reduced inflammation. It also modulated gut microbiota by increasing beneficial bacteria and enhancing mucin production (MUC2 and MUC3), which strengthen gut barrier function.
Section 3.3.4 Cancer, Page 14, Line: 403-414
- plantarum IS-10506 is a probiotic strain isolated from Indonesian fermented buffalo milk (dadih) with demonstrated gut barrier-enhancing properties. It has been shown to mitigate intestinal epithelial damage by inhibiting nuclear factor kappa B (NF-κB) activation and downregulating tumor necrosis factor receptor-1 (TNFR1), which play key roles in inflammatory responses. It significantly reduced blood lipopolysaccharide (LPS) levels, indicating reduced microbial translocation and improved gut integrity in HIV-infected children undergoing antiretroviral therapy (ARV). However, its effect on systemic and humoral immune responses, including CD4+/CD8+ T cell ratios and fecal secretory IgA (sIgA) levels, was not significant. It increased the number of regulatory T cells [59] in children receiving first-line antiretroviral therapy (ART), suggesting a potential role in immune regulation for managing HIV-related immune dysregulation.
Section 3.3.5 Human Immunodeficiency Virus (HIV), Page 14, Line: 417-429
- plantarum LB931 [60] was observed to maintain a low vaginal pH, which helps prevent harmful microbial overgrowth. This contributes to reducing the risk of bacterial vaginosis and yeast infections, promoting overall vaginal health. It helps maintain a low vaginal pH, which is crucial for preventing harmful microbial overgrowth. The high numbers of lactobacilli, including LB931, had significantly lower vaginal pH levels, which is associated with a reduced prevalence of Group B streptococci (GBS) and yeast infections.
Section 3.3.6 Vaginal Health, Page 14, Line: 431-436
- plantarum 299v is a well-studied probiotic strain recognized for its resilience in the gastrointestinal tract, surviving gastric acidity and bile exposure. It has demonstrated antimicrobial properties, reducing bacterial translocation and systemic inflammation, particularly in critically ill patients. Despite its ability to attenuate systemic inflammation and support gut microbiota restoration, the study found no significant results.
Section 3.3.7 ICU Patients, Page 15, Line:441-446
- plantarum PS128 was also linked to improvements on the SNAP-IV scale, a widely used measure for ADHD and ASD symptoms, suggesting its potential adjunctive role in cognitive and behavioral therapy for ASD patients. L. plantarum PS128 has ability to modulate gut microbiota and influence neurotransmitter levels, such as dopamine and serotonin, in the brain. Additionally, the probiotic was well-tolerated, with fewer side effects
Section 3.4.1 Autistic Spectrum Disorders (ASD), Page 15, Line:450-455
- plantarum PS128 [70] did not exhibit tic-reducing effects in children with Tourette syndrome. Tourette syndrome results from a complex interaction between social–environmental factors, multiple genetic abnormalities, and neurotransmitter disturbances. The present study reflected that intervention with PS128 did not have a superior response in tic severity improvement. There are a few possible explanations. L. plantarum PS128 has been demonstrated in germ-free mice to increase concentrations of dopamine, serotonin, and their metabolites in the striatum. However, it is known that patients with Tourette syndrome may have dopamine hypersensitivity and, therefore, respond to dopamine blocking agents such as risperidone, pimozide, and haloperidol. Therefore, it may be explainable that the use of PS128 did not show a superior response in improving tic severity. This suggests that while L. plantarum PS128 may be beneficial for ASD, its therapeutic benefits may not extend to all neurological or psychological conditions.
Section 3.4.2 Tourette Syndrome, Page 15, Line:457-468
- plantarum HEAL9 [52] significantly reduced acute and chronic stress, indicating its stress-relieving potential. It showed a significant reduction in awakening cortisol levels, a key marker of stress response. Additionally, improvements were observed in mood, particularly in reducing confusion, anger, and depressive symptoms. The probiotic also enhanced cognitive function, particularly in memory and learning tasks, suggesting its role in modulating the gut-brain axis to alleviate stress and improve mental well-being.
Section 3.4.3 Psychological and Cognitive Function, Page 15, Line:471-477
In elderly populations, L. plantarum OLL2712 [61] demonstrated memory function decline-protective effects, suggesting its potential in maintaining cognitive health in aging individuals by modulating neuroinflammation through the microbiota-gut-brain axis. The probiotic’s high interleukin-10 (IL-10)-inducing activity helped suppress neuroinflammation by inhibiting pro-inflammatory cytokines such as tumor necrosis factor-alpha (TNF-α) and interleukin-1β (IL-1β). Then L. planatrum OLL2712’s immunomodulatory properties may help mitigate age-related cognitive decline and neurodegenerative processes.
Section 3.4.4 Cognitive Health and Aging, Page 16, Line: 487-494
- plantarum PS128 showed significant decreases in Hamilton Depression Rating Scale-17 (HAMD-17) and Depression and Somatic Symptoms Scale (DSSS) scores, indicating improved mood and reduced somatic symptoms. It also influence neurotransmitter pathways and gut permeability markers, such as zonulin and intestinal fatty acid-binding protein (I-FABP), suggests a role in regulating systemic inflammation and neuroimmune interactions in MDD.
Section 3.4.5 Depression and Sleep Regulation, Page 16, Line: 498-503
- plantarum GMNL6 significantly decreased melanin synthesis, potentially through the inhibition of the mitogen-activated protein kinase (MAPK) signaling pathway and the activation of protein kinase B (AKT), which suppresses melanogenesis. Additionally, lipoteichoic acid (LTA) from L. plantarum GMNL6 was identified as a key component contributing to its skin-lightening effects. A clinical trial further confirmed that a heat-killed L. plantarum GMNL6 cream significantly reduced melanin index (M index).
Section 3.5.1 Skin Health, Page 16, Line: 512-518
It effectively prevented and treated early childhood caries (ECC) by modulating oral microbiota and reducing pathogens. L. plantarum CCFM8724 also significantly decreased Streptococcus mutans and Candida albicans while increasing beneficial genera like Granulicatella and Gemella. The probiotic also reduced caries-associated bacteria, helping restore microbial balance and prevent biofilm formation, making it a promising ECC intervention.
Section 3.5.2 Oral Health, Page 17, Line: 529-534
- plantarum K50 altered gut microbiota by increasing Lactobacillus plantarum and reducing Actinobacteria, changes correlated with visceral adiposity modulation. These findings suggest L. plantarum K50's potential as a microbiome-targeted intervention for dyslipidemia management.
Section 3.6.1 Lipid Profile, Page 17, Line: 548-552
Despite no significant changes in fasting plasma glucose (FPG), homeostasis model assessment of insulin resistance (HOMA-IR), and the quantitative insulin sensitivity check index (QUICKI), L. plantarum HAC01 may exert its effects via gut microbiota modulation rather than direct insulin sensitization.
Section 3.6.2 Blood Glucose Levels, Page 17, Line 559-562
The probiotic also reduced serum insulin levels, homeostasis model assessment of insulin resistance (HOMA-IR), and leptin levels, indicating improved metabolic regulation. Additionally, gut microbiota analysis revealed increased Lactobacillus and Oscillibacter levels which the latter being negatively correlated with triglyceride and alanine transaminase (ALT) levels, suggesting a role in lipid metabolism and adiposity control. These findings support L. plantarum LMT1-48’s therapeutic potential as a microbiome-targeted intervention for obesity management.
Section 3.6.3 Anti-Obesity, Page 17-18, Line:570-576
The probiotic also improved fatigue-associated biomarkers, reducing serum lactate and ammonia accumulation during exercise and recovery phases. Furthermore, L. plantarum TWK10 supplementation led to favorable body composition changes, including reduced fat mass and increased muscle mass. These effects may be linked to enhanced short-chain fatty acid (SCFA) production and gut microbiota modulation, suggesting L. plantarum TWK10 as a promising ergogenic aid.
Section 3.6.4 Physiological Adaptations, Page 18, Line: 586-591
- The lack of appropriate table describing the results also persist.
Response: A new Table of Results and concise describing the table and figure were added into the studied. The table 2: Table of Results: Summarized the L. plantarum’s strains therapeutic potential on various health conditions (from Table S.1 Data Characteristics Table in the Supplementary Materials). The table provide a clear and extensive list of various L. plantarum strains with human’s health conditions.
However, the table 1, the overall classification by symptoms with reference number were also summarized. For gastrointestinal system, there were classified into 5 major symptoms such as IBS symptoms, bowel functions, microbiome diversity, diarrhea and constipation. For IBS symptoms, there are 7 studies included in this study. There were classified into 7 major symptoms for immune system, 3 symptoms for central nervous system, 2 symptoms for integumentary system and 3 symptoms for miscellaneous group.
Section 3. Results, Page 6, Line 235-240 (Describe of Table 1)
The graphical of L.plantarum strains classified by symptoms also shown in the Fig. 2. The Sun Burst graphical clearly presents the classification of L. plantarum strains by symproms. From the Fig 2., for infection disease under the subset of immune system, there are several L.plantarum strains help alleviates infection diseases such as L. plantarum P17630 (2 studies), L. plantarum DR7 (2 studies), L. plantarum I1001 and L. plantarum PCS26. Therefore, this graphical will significantly help the readers to clearly understand the overall studied in a glance.
Section 3. Results, Page 6, Line 241-247. (Describe of Figure 2, graphical)
The table of results also shown in the Table 2 which summarized from the Data Characteristics Table, Table S.1 in the Supplementary Materials. From the Table 2, L. plantarum CCFM1143 helps release Chronic diarrhea, while L. plantarum DR7 helps reduced anxiety, improved cognition, and URTI symptoms. However, L. plantarum 299v resulted in mixed results, some studies shown positive and some shown negative effects. There may be several related factors, but confound factors may play a major role on this issue such as diet, age, genetic differences, and gut microbiome diversity, which may change the response to treatment, also have not been studied in this research.
Section 3. Results, Page 6, Line between 248-255 (Describe of Table 2.)
Reviewer 4 Report
Comments and Suggestions for Authors
Peer Review Report
Research Focus and Significance
The study systematically reviews strain-specific effects of Lactiplantibacillus plantarum, highlighting its therapeutic potential. It is relevant due to the increasing interest in probiotics and their clinical use. While it effectively synthesizes existing research, a clearer distinction of its novelty compared to previous reviews would enhance its contribution.
Originality and Contribution
The manuscript compiles extensive information on L. plantarum strains, but its novelty is unclear. Clarifying how it advances the field—by identifying knowledge gaps or proposing new clinical applications—would enhance its impact. The authors should highlight the unique insights gained compared to previous studies.
Methodological Rigor
The adherence to PRISMA-ScR guidelines and the PICO framework enhances methodological robustness; however, some areas need further clarification. Regarding the search strategy, while PubMed and Embase were used, it is unclear whether additional databases like Cochrane or Scopus were considered. If not, a justification for this limitation is necessary. The study inclusion and exclusion criteria require a more detailed explanation of how study quality was ensured and potential biases were addressed. Additionally, the manuscript does not discuss inter-rater reliability among the four independent reviewers or whether discrepancies were systematically resolved.
Validity of Conclusions
The conclusions are consistent with the presented results, but a more nuanced discussion on strain-specific variability would enhance the study. Some strains show inconsistent or conflicting results across different studies, and exploring potential reasons for this would strengthen the conclusions. Additionally, certain claims about clinical applications lack a critical evaluation of study limitations. The review would also benefit from addressing potential confounding factors affecting strain efficacy, such as differences in the host microbiome or dietary influences.
References and Data Presentation
The references are comprehensive and well-cited, but some areas need improvement. The manuscript relies heavily on recent studies, and incorporating foundational works on L. plantarum would provide a stronger contextual basis. Certain sections would benefit from summary tables or graphical representations to improve readability. Additionally, figures and tables should be more explicitly cross-referenced within the text.
Final Recommendation
The manuscript offers valuable insights into the therapeutic applications of L. plantarum, but refinements are needed to enhance clarity and impact. It would benefit from a clearer distinction of its unique contributions compared to previous literature, as well as greater methodological transparency, particularly in database selection, quality control, and inter-rater reliability. Expanding the discussion on strain-specific inconsistencies and potential confounding factors would strengthen the analysis. Additionally, improving data presentation with more figures or tables would enhance visualization.
Final Decision: Minor Revisions Required
With these improvements, the manuscript has strong potential for publication. I look forward to reviewing the revised version.
Comments on the Quality of English LanguageThe English could be improved to more clearly express the research
Author Response
Response to Reviewer No. 4
- The English could be improved to more clearly express the research.
Response: The authors try to revise some unclear English expressions to be clearer throughout the manuscript.
- Does the Introduction provide sufficient background and included all relevant references? (Can be improved)
Response: The Introduction section was added a paragraph to discuss about the background of L. plantarum and also included some more relevant references.
Section 1. Introduction section, Page: 2, Line: 43-50.
- Is the research design appropriate? (Yes)
Response: Yes
- Are the methods adequately described? (Can be improved)
Response: Please see the details response in item No. 10 (Methodological Rigor)
- Are the results clearly presented? (Can be improved)
Response: The Results section was added some discussion of Table1. Lactiplantibacillus plantarum’s classification by symptoms with reference number, Figure 2. L.plantarum strains classified by symptoms and Table of results, Table 2 as follows:
However, The table 1, the overall classification by symptoms with refernce number were also summarized. For gastrointestinal system, there were classified into 5 major symtoms such as IBS symptoms, bowel fundctions, microbiome diversity, diarrhea and constipation. For IBS symptoms, there are 7 studies included in this study. There were classifed into 7 major symtoms for immune system, 3 symptoms for central nervous system, 2 symptoms for integumentary system and 3 symptoms for miscellaneous group
Section 3. Results: Page 6, Line: 235-240
The graphical of L.plantarum strains classified by symptoms also shown in the Fig. 2. The Sun Burst graphical clearly presents the classification of L. plantarum strains by symproms. From the Fig 2., for infection disease under the subset of immune system, there are several L.plantarum strains help alleviates infection diseases such as L. plantarum P17630 (2 studies), L. plantarum DR7 (2 studies), L. plantarum I1001 and L. plantarum PCS26. Therefore, this graphical will significantly help the readers to clearly understand the overall studied in a glance.
Section 3. Results: Page 6, Line: 241-247
The table of results also shown in the Table 2 which summarized from the Data Characteristics Table, Table S.1 in the Supplementary Materials. From the Table 2, L. plantarum CCFM1143 helps release Chronic diarrhea, while L. plantarum DR7 helps reduced anxiety, improved cognition, and URTI symptoms. However, L. plantarum 299v resulted in mixed results, some studies shown positive and some shown negative effects. There may be several related factors, but confound factors may play a major role on this issue such as diet, age, genetic differences, and gut microbiome diversity, which may change the response to treatment, also have not been studied in this research.
Section 3. Results: Page 6, Line: 248-25
- Are the conclusions supported by the results? (Can be improved)
Response: Please see the details response in item No. 11 (Validity of Conclusions)
- The lack of appropriate table describing the results also persist.
Response: A new table of Results was added to be Table 2., in the manuscript. This table explicitly shown the summarized of each L. plantarum strains with the humans’ health conditions.
Section 3. Results, (Table 2.) page 7-11, line between 260-261
- Research Focus and Significant:
- While it effectively synthesizes existing research, a clearer distinction of its novelty compared to previous reviews would enhance its contribution.
Response: In summary, most of the studies reviewed utilized only a single strain of Lactobacillus plantarum as a therapeutic agent to improve specific aspects of human health and well-being. However, the present study uniquely compiles multiple individual strains of L. plantarum and their diverse health applications for humans into one comprehensive overview.
Section 5. Conclusions, page 20, Line: 681-685.
- Originally and Contribution:
- The manuscript complies extensive information on plantarum strains, but its novelty is unclear. Clarify how it advances the field-by identifying knowledge gaps or proposing new clinical applications-would enhance its impact. The authors should highlight the unique insights gained compared to previous studies.
Response: The periodic advances in research are expected to include microbiome profiling, precision probiotic interventions, and probiotic multi-strain synergies to maximize efficacy in therapy. The more direct treatment may be considered to use, in the future, such as Fecal Microbiota Transplantation (FMT), may represent a promising advanced strategy to restore gut microbiome and enhance the cognitive functions of healthy people and patients with neurological disorders. With the current and continuing expansion of probiotic science, L. plantarum will surely have the spotlight in the personalized, enhanced in the new advanced technology and evidence-based probiotic therapies that mainstream healthcare will be using in the future neurological disorders. With the current and continuing expansion of probiotic science, L. plantarum will surely have the spotlight in the personalized, enhanced in the new advanced technology and evidence-based probiotic therapies that mainstream healthcare will be using in the future.
Section 5. Conclusions, Page 20, Line: 686-694
- Methodological Rigor:
- The adherence to PRISMA-ScR guidelines and the PICO framework enhances methodological robustness; however, some areas need further clarification. Regarding the search strategy, while PubMed and Embase were used, it is unclear whether additional databases like Cochrane or Scopus were considered. If not, a justification for this limitation is necessary.
Response:
In fact, it was our limitation to review the paper from all four databases in this study, Scopus, Embase, PubMed and Cochrane library. In our study, the search strategy used the papers from Embase and PubMed database because they are very well-known and well accepted databases in biomedical fields. Therefore, we decided to use only these two databases for our studies.
Section 2.7 Methods, Research Design and PRISMA Flow Diagram, page 4, Line : 152-156
- The study inclusion and exclusion criteria require a more detailed explanation of how study quality was ensured and potential biases were addressed.
Response:
All relevant text, tables, and figures in the included studies were reviewed for data extraction. The following types of studies were excluded: (1) Non-original research (e.g. reviews, protocols, letters, comments, and guidelines); (2) Studies that did not focus on L. plantarum monotherapy (i.e., mixing with other probiotic species); (3) Non-human studies; (4) Unpublished, grey literature or non-peer-reviewed studies; and (5) Studies published in languages other than English in order to avoid any misunderstanding or language bias may concerned.
Section 2.4 Data Extraction, Page 3, Line 126-132.
The inclusion criteria were established to select high-quality studies reporting primary data on human clinical outcomes related to specific L. plantarum strains. Following a systematic screening and selection process, data were extracted and qualitatively synthesized. Results were categorized by targeted health conditions, enabling a thorough assessment of therapeutic efficacy.
Strict inclusion and exclusion criteria ensured the selection of scientifically rigorous studies focused explicitly on distinct L. plantarum strains and their strain-specific clinical outcomes. This strategy minimized bias and enhanced the reliability of findings, supporting evidence-based clinical applications and future research.
Section 2.7 Methods, Methods, Research Design and PRISMA Flow Diagram, Page 4, Line 157-165.
- Additionally, the manuscript does not discuss inter-rater reliability among the four independent reviewers or whether discrepancies were systematically resolved.
Response:
Inter-rater reliability was assessed by having multiple reviewers independently screen studies based on predefined criteria. Discrepancies were resolved through consensus discussions or consultation with a third reviewer. This method ensured consistency, reduced subjective bias, and strengthened methodological integrity.
Section 2.7 Methods, Methods, Research Design and PRISMA Flow Diagram, Page 4, Line 166-169.
- Validity of Conclusions:
- The conclusions are consistent with the presented results, but a more nuanced discussion on strain-variability would enhance the study.
Response: Some of the strain-variabilities were already discussed in the Conclusion section.
Highlighting the therapeutic versatility of L. plantarum strains across different physiological systems, it also observed how their inefficacies are expressed strain-specifically in establishing gastrointestinal health, immune modulation, central nervous system disorders, metabolic regulation, skin health, and physical performance. For example, irritable bowel syndrome, diarrhea, constipation, and bowel dysfunction symptoms have been observed to have improved efficiently by strains such as L. plantarum Lpla33 and L. plantarum 299v, perhaps as a result of mechanisms involving gut barrier enhancement and inflammation modulation. Whereas L. plantarum IS-10506 increases mucosal immunity, L. plantarum PS128 seems to operate either via the gut-brain axis or closer peripheral sites to change certain neuropsychological functions. Other strains prove beneficial for improving metabolic health; for instance, L. plantarum EGCG 13110402 and L. plantarum HAC01 are implicated in downregulating lipid metabolism and reducing blood glucose levels. These findings empirically suggest that L. plantarum strains can be possible significant treatment adjuncts, particularly for applications in personalized medicine and integrative healthcare.
Section 5. Conclusion, Page 19, Line: 638-652
- Some strains show inconsistent or conflicting result across different studies, and exploring potential reasons for this would strengthen the conclusion.
Response:
The potential confounding factors affecting strain efficacy, such as differences in the host microbiome or dietary influences. Therefore, some L. plantarum strains shown inconsistency results such as L. plantarum 299v shown mixed results, some studies shown positive and some negative effects. There may be several related factors, but confound factors may play a major role on this issue.
Section 5. Conclusions, Page 19, Line 659-663.
- Additionally, certain claims about clinical applications lack a critical evaluation of study limitations.
Response:
While this study applied a rigorous and systematic methodology to evaluate the clinical applications of L. plantarum strains, it is acknowledged that certain claims may still lack a critical evaluation due to inherent limitations in the included studies. The decision to restrict the literature search to Embase and PubMed, though justified by their strong biomedical relevance. Additionally, while stringent inclusion criteria were applied to ensure high-quality evidence, variations in study design, sample size, and clinical endpoints among the included studies could impact the strength of conclusions drawn. To mitigate these limitations, a cautious and evidence-based interpretation of findings was adopted, highlighting the need for further well-controlled clinical trials to validate the therapeutic potential of L. plantarum strains.
Section 5. Conclusions, Page 19, Line: 671-680.
- The review would also benefit from addressing potential confounding factors affecting strain efficacy, such as differences in the host microbiome or dietary influences.
Response:
Confounding variables like diet, age, sex, genetic differences, and gut microbiome diversity, which can change the response to treatment, also have not been studied in this research. The potential confounding factors affecting strain efficacy, such as differences in the host microbiome or dietary influences. Therefore, some L. plantarum strains shown inconsistency results such as L. plantarum 299v shown mixed results, some studies shown positive and some negative effects. There may be several related factors, but confound factors may play a major role on this issue.
Section 5. Conclusions, Page 19, Line: 657-663
- References and Data Presentation:
- The references are comprehensive and well-cited, but some areas need improvement. The manuscript relies on recent studies and incorporating foundation works on L. plantarum would provide a stronger contextual basis. Certain sections would benefit from summary tables or graphical representations to improve readability. Additionally, figures and tables should be more explicitly cross-referenced within the text.
Response: A new table of result, summarized from the Table S.1 Data Characteristics Table from the supplementary materials, was added in the Results section as Table 2.
Section 3. Results, (Table 2.) page 7-11, line between 260-261
Round 3
Reviewer 3 Report
Comments and Suggestions for Authors
Although my initial impression was that the manuscript does not contain the necessary degree of innovation (and to some extent I still believe it does), I have to admit that the manuscript has improved a lot during the revision process, and that it is well-written and in accordance with the most commonly used methodology. I will not object to its acceptance in the current version.